# Functional in vivo characterization of *sox10* enhancers in neural crest and melanoma development

Rebecca L. Cunningham [1], Eva T. Kramer [1], Sophia K. DeGeorgia [1], Paula M. Godoy[1], Anna P. Zarov [1], Shayana Seneviratne [2], Vadim Grigura[1] & Charles K. Kaufman [1✉]

The role of a neural crest developmental transcriptional program, which critically involves Sox10 upregulation, is a key conserved aspect of melanoma initiation in both humans and zebrafish, yet transcriptional regulation of *sox10* expression is incompletely understood. Here we used ATAC-Seq analysis of multiple zebrafish melanoma tumors to identify recurrently open chromatin domains as putative melanoma-specific *sox10* enhancers. Screening in vivo with EGFP reporter constructs revealed 9 of 11 putative *sox10* enhancers with embryonic activity in zebrafish. Focusing on the most active enhancer region in melanoma, we identified a region 23 kilobases upstream of *sox10*, termed *peak5*, that drives *EGFP* reporter expression in a subset of neural crest cells, Kolmer-Agduhr neurons, and early melanoma patches and tumors with high specificity. A ~200 base pair region, conserved in *Cyprinidae*, within *peak5* is required for transgenic reporter activity in neural crest and melanoma. This region contains dimeric SoxE/Sox10 dimeric binding sites essential for *peak5* neural crest and melanoma activity. We show that deletion of the endogenous *peak5* conserved genomic locus decreases embryonic *sox10* expression and disrupts adult stripe patterning in our melanoma model background. Our work demonstrates the power of linking developmental and cancer models to better understand neural crest identity in melanoma.

[1] Division of Medical Oncology, Department of Medicine and Department of Developmental Biology, Washington University in Saint Louis, St. Louis, MO, USA. [2] School of Arts and Sciences, Washington University in Saint Louis, St. Louis, MO, USA. ✉email: ckkaufman@wustl.edu

Melanoma is a potentially deadly cancer of transformed melanocytes, which are pigment-producing cells derived from embryonic neural crest. Early detection and treatment of melanoma is critical because of high mortality from metastatic melanoma and the potential for long-term side effects of the most effective immunotherapy treatments. The difficulty of treating metastatic melanoma underscores the necessity of defining how this cancer initiates and the urgency of developing new therapeutics[1]. Encouragingly, chromatin modifiers and other proteins that shape the epigenetic landscape offer promising novel therapeutic targets that may expand the treatment repertoire beyond immunotherapy and MAPK-targeting strategies[2,3] and combat the ability of melanomas to evade treatment due to their high mutational burden, heterogeneous mutations, and rapid tumor evolution[4,5].

To effectively treat melanoma in its earliest stages requires a thorough understanding of mechanisms that drive melanoma initiation. The transgenic $BRAF^{V600E}/p53^{-/-}$ zebrafish has been established as a powerful melanoma model that is highly reflective of the human disease and can be utilized to study early stages of melanoma initiation[6]. In this model, the human $BRAF^{V600E}$ oncogene is expressed under the control of the *mitfa* promoter, a melanocyte-specific promoter. Within $BRAF^{V600E}/p53^{-/-}$ zebrafish, all melanocytes have the potential to become melanoma, but only one to three melanomas typically develop from the many thousands of melanocytes in any single fish, presenting an excellent model in which to study how cancer initiates from a field of cancer-prone cells.

Recent work in zebrafish has argued that reactivation of a neural crest program (NCP) is an early step in melanoma initiation[7]. A single melanoma cell can be visualized in live zebrafish by expression of the *crestin:EGFP* transgene[7]. Not only does *crestin:EGFP* expression uniquely enable visualization of cancer before a large tumor is visible, expression of *crestin*, which is absent in premalignant melanocytes and becomes evident in the first detectable cells of melanoma, indicates a re-emergence of aspects of neural crest transcriptional identity in these cells because *crestin* is exclusively expressed in the embryonic neural crest of zebrafish[8,9]. Neural crest cells (NCCs) are a transient and migratory embryonic cell population that give rise to a variety of cell types, including melanocytes. Prior studies have noted that NCCs and cancer cells share many cell biological characteristics, including the ability to proliferate, change cell adhesion properties, and migrate[10,11]. For example, epithelial-to-mesenchymal transition occurs to permit both NCC migration and cancer metastasis[12]. Furthermore, melanoma cells can exhibit neural crest-like behavior after transplantation into chick embryos, demonstrating their plasticity and connection to their developmental origins[13].

While *crestin* is expressed in the neural crest, this gene is zebrafish-specific and functionally uncharacterized; therefore, *crestin* itself does not represent a future therapeutic target in human melanoma. However, *crestin* is one of a suite of neural crest genes that are expressed in melanoma in zebrafish and humans[7]. The re-emergence of an NCP transcriptional program in melanoma is supported by the upregulation of other neural crest-related genes in tumors such as *sox10*, *dlx2a*, *ednrb*, *Brn3*, and *FOXD1*[7,14–17]. Applying the knowledge of developmental mechanisms in NCCs may enhance our understanding of melanoma initiation and treatment of NCC-derived cancers. Therefore, we aim to identify neural crest regulatory elements that not only serve as a marker of melanoma, such as *crestin*, but also regulate *sox10*, which is conserved across species, to understand the foundations of transcriptional control of genes contributing to melanoma initiation.

One highly upregulated neural crest gene in melanoma in both zebrafish and humans is *sox10*, a transcription factor necessary for neural crest development[7,18–20]. Genetic manipulation of *sox10* activity in a zebrafish $BRAF^{V600E}$ melanoma model affects melanoma onset[7], and abrogation of *sox10* affects melanoma maintenance in a *Nras* mouse melanoma model and in human melanoma cell culture[21]. Likewise, overexpression of *sox10* in melanocytes increases the rate of melanoma onset in zebrafish[7]. Further supporting a role for a NCP in melanoma, regulatory elements near neural crest genes, such as *sox10*, are activated in melanoma cell lines and NCCs, but not generally in other adult tissues or a range of cancer cell lines. Importantly, this gene expression program phenomenon is conserved in human melanomas[7].

Since *sox10* is critical to establish neural crest identity, the development of melanocytes, and the maintenance and progression of melanoma[7,21,22], transcriptional regulation of *sox10* may be critical for re-establishing or enhancing neural crest identity in melanoma initiation. Therefore, identifying and functionally characterizing enhancer elements that control *sox10* expression during this oncogenic transition will broaden our understanding of transcriptional and, eventually, epigenetic changes that occur to trigger melanoma initiation and activation of an NCP-like identity from within a cancerized field.

In this study, we functionally screen putative enhancer elements, identified as accessible or "open" regions of chromatin using ATAC-Seq on zebrafish melanomas, near the *sox10* gene for transcriptional activating, or enhancer, activity in embryonic neural crest as well as in melanoma tumors. This screening approach allowed us to identify and focus initially on one 669 base pair (bp) enhancer, termed *peak5*, that is highly and specifically active in adult zebrafish melanoma. We further show that *peak5* is an active embryonic enhancer in a subset of the NCCs and a select population of Kolmer–Agduhr neurons in the central nervous system. We also identify a ~200 bp nucleotide sequence centered within *peak5* that is conserved between members of the fish *Cyprinidae* family and is necessary and sufficient for *peak5* activity in a subset of NCCs, as well as necessary for activity in melanoma. Furthermore, mutational analysis of a dimeric SoxE binding site in the conserved core of *peak5* uncovers transcription factor binding sites that are necessary for robust *peak5* neural crest and melanoma activity. Excitingly, in vivo deletion of the endogenous *peak5* conserved locus results in decreased embryonic *sox10* expression. Adult *peak5* homozygous mutants also display a disruption of stripe patterning, further demonstrating that *peak5* is a functional enhancer. Together, these data expand our understanding of transcriptional regulatory changes that control aspects of neural crest programs during normal development and in melanoma, and suggest a model wherein autoregulation of *sox10* in a feed-forward loop may contribute to the formation of melanoma.

## Results

**Identification of putative *sox10* enhancers**. To identify putative regulatory elements surrounding the *sox10* locus in melanoma, ATAC-Seq was performed on bulk melanoma tumors isolated from $Tg(BRAF^{V600E};crestin:EGFP);p53^{-/-}$ zebrafish (Fig. 1a). ATAC-Seq on melanoma tumor cells showed consistent regions of open chromatin and identified 11 such regions of open chromatin around and within the *sox10* locus, including a putative *sox10* minimal promoter encompassing exon 1 of *sox10* (Fig. 1b; Supplementary Fig. 1a). These open regions of chromatin were also consistent with ATAC-Seq peaks in zebrafish melanoma cell lines[7] (Supplementary Fig. 1a), as well as ATAC-Seq peaks in zebrafish embryonic neural crest by the 5-6 somite stage (Supplementary Fig. 1b)[23–26]. To assess if these ATAC-Seq peaks are functional enhancers in neural crest and melanoma cells, we

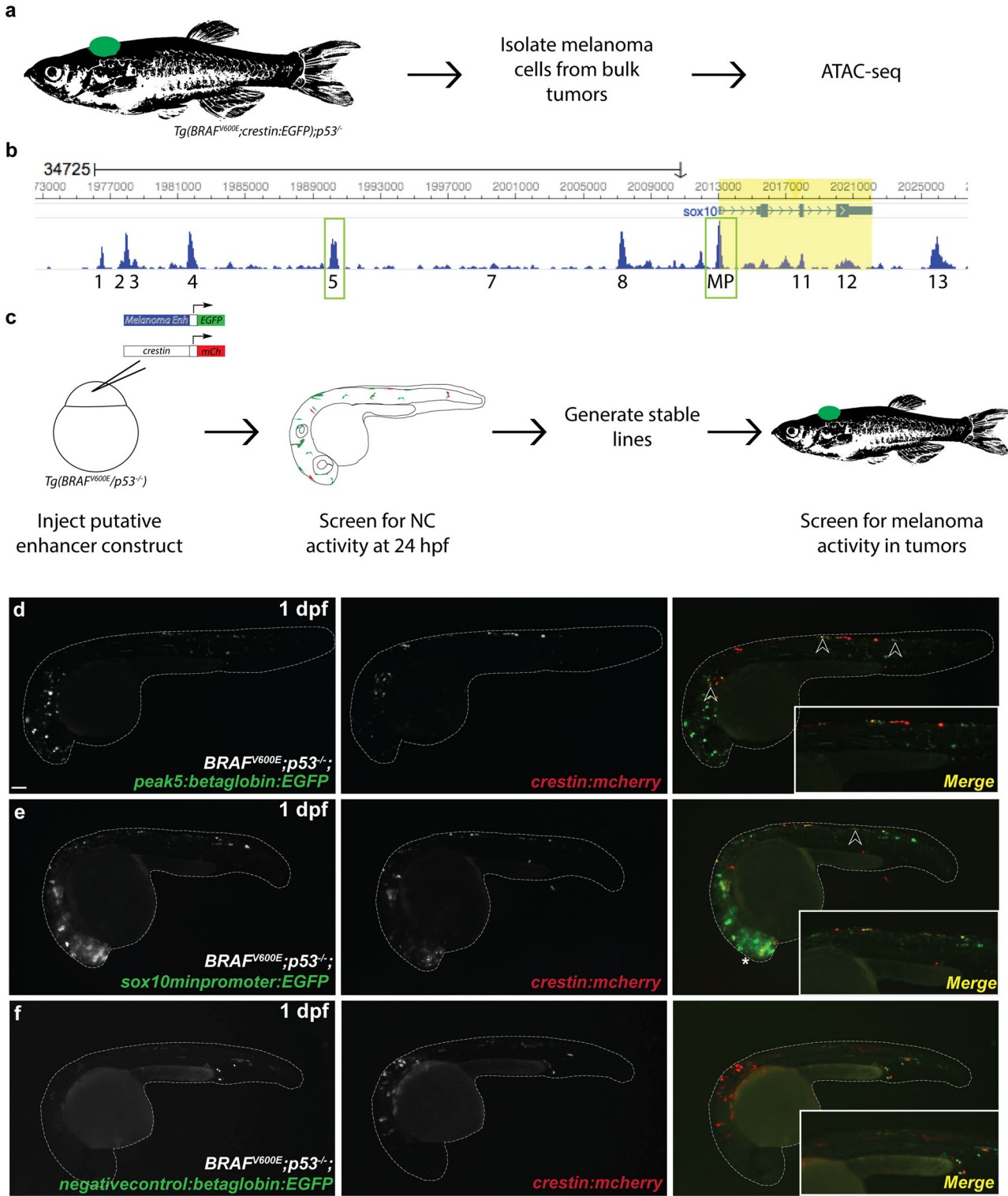

**Fig. 1 ATAC-Seq of zebrafish melanoma tumors identifies *peak5* as an active neural crest enhancer. a** Schematic for obtaining zebrafish melanoma tumor cells for ATAC-Seq. **b** Regions of open chromatin surrounding the *sox10* locus. Peak numbers are annotated below. MP minimal promoter. **c** Schematic for screening putative enhancers for activity in NCCs and melanoma. **d** *peak5*-driven *EGFP* expression is mosaically present in NCCs (arrowheads) in F0 injected embryos, as indicated by co-localization with *crestin:mch* at 1 dpf. **e** The *sox10* minimal promoter in F0 embryos is active in both NCCs (arrowheads) and the CNS (asterisk). **f** F0 embryos injected with a negative control (~76 kb upstream of the *sox10* TSS) exhibit only limited expression of *EGFP* not localized to NCCs. Insets show enlarged region above yolk extension.

performed in vivo reporter assays in $Tg(BRAF^{V600E});p53^{-/-}$ zebrafish embryos (Fig. 1c). Putative non-intronic enhancers that were present in at least 4 out of 5 tumor ATAC-Seq samples were cloned upstream of the mouse *beta-globin* basal promoter (an element previously shown to have minimal detectable intrinsic cell-type-specific activity) driving *EGFP*[27], within a Tol2 vector (plasmid 394 from the Tol2 Kit[28]) to enable mosaic integration into the zebrafish genome. Each putative enhancer construct was injected into single-cell $Tg(BRAF^{V600E});p53^{-/-}$ embryos. A *crestin:mCherry* construct was also co-injected with each putative enhancer construct to serve as a positive injection control and to co-label NCCs. Embryos were screened at 1 day post fertilization (dpf) for *EGFP* expression and co-labeling of EGFP and mCh, indicating that the enhancer functions in the neural crest. Larva was ranked as either "no expression," "weak expression," or "strong expression" for both *EGFP* and *mCherry*. "Strong expression" was characterized by 5 or more EGFP positive cells whereas "weak expression" was characterized by fewer than 5 EGFP positive cells. Nine out of eleven putative enhancers drove *EGFP* expression in at least a subset of NCCs, as determined by cell shape, location, and/or co-labeling with *crestin:mCherry* (Supplementary Fig. 2).

One peak of interest, termed *peak5*, cloned as a 669 bp element (Supplementary Fig. 3), exhibited particularly robust activity in NCCs (Fig. 1d). The putative *sox10* minimal promoter (*sox10-min*), cloned as a 351 bp element and likely positive control, exhibited more robust *EGFP* expression throughout the nervous system and in NCCs (Fig. 1e). A region of closed chromatin, as determined by ATAC-Seq, approximately 76 kb upstream of *sox10* was also cloned upstream of *betaglobin:EGFP* as a negative control (Negative Control B) and did not drive neural crest *EGFP* expression (Fig. 1f). In subsequent analyses, we noted that this control region also exhibits sequence similarity to another region on Chromosome 3 (Chr3: 45240516-45241216). An additional negative control (Negative Control A) cloned from a region 10.7 KB upstream of *sox10* also exhibited minimal reporter activity (Supplementary Fig. 2). Collectively, ATAC-Seq analysis of zebrafish melanoma tumors reliably identifies transcriptional regulatory regions near *sox10* that function as enhancers with neural crest cell activity in transgenic reporter studies.

**peak5 is active in embryonic neural crest and Kolmer–Agduhr neurons**. Enhancer reporter constructs injected into zebrafish embryos are randomly integrated throughout the genome, and thus subject to position effects[29]. In addition to examining numerous F0 transgenic embryos for consistent spatial and temporal expression as above, we further characterized the expression pattern of activity of *peak5*, a region showing particularly robust neural crest activity in zebrafish embryos, by generating six independent stable transgenic lines. To generate stable lines, F0 injected embryos were raised to maturity and then outcrossed to $Tg(BRAF^{V600E});p53^{-/-}$ or $Tg(sox10(7.2):mRFP)$[30] zebrafish. Subsequent F1 embryos were then screened for *EGFP* expression at 1 dpf. All 6 identified stable transgenic lines exhibited commonalities in expression patterns, detailed below (Fig. 2a–c; Supplementary Fig. 4a-b).

At 1 dpf, 4 out of 6 stable lines exhibited EGFP expression in premigratory NCCs dorsally located on the posterior trunk (Fig. 2a, b; Supplementary Fig. 4a-b). To confirm that *peak5* is active in NCCs and cells that would typically express *sox10* we crossed $Tg(peak5:betaglobin:EGFP)$, in which EGFP is localized cytoplasmically, to the previously published $Tg(sox10(7.2):mRFP)$ line, in which mRFP is membrane localized[30]. Lightsheet imaging of double transgenic embryos at 1 dpf in both dorsal (Fig. 2d–d″) and lateral views (Fig. 2e–e″) showed most EGFP positive cells in

the dorsal posterior region co-localize with *sox10:mRFP* expressing cells at 1 dpf (Fig. 2d″, e″), illustrating that *peak5* is active in NCCs as labeled by a *sox10*-driven reporter. Additional epifluorescence imaging of $Tg(peak5:betaglobin:EGFP);Tg(crestin:mCherry)$ 1 dpf embryos shows similar co-localization of EGFP and mCherry (Supplementary Fig. 4d-d″). In addition to a subset of NCCs, 6 out 6 *peak5* transgenic lines express *EGFP* within a subset of ventral Kolmer–Agdhur (KA) neurons in the spinal cord, which contact cerebrospinal-fluid, as identified by cell shape and localization (Fig. 2a–c; Supplementary Fig. 4a-c, e). By 5 dpf, *EGFP* expression was consistently observed in KA neurons across all lines, and in the heart, a structure containing NCC derivatives, in 3 out 6 stable lines (Fig. 2b; Supplementary Fig. 4b-c). Only one stable line, *peak5_115A*, exhibited EGFP signal in muscle, in addition to the reproducible expression seen in KA neurons and the heart as seen in the other lines (Supplementary Fig. 4c). Given that *sox10* is not endogenously expressed in muscle, we reasoned that this muscle activity was due to position effects or enhancer trapping of the transgene while the reproducible expression in the KA neurons of the CNS and the heart in multiple lines is more likely an accurate read-out of these enhancer domains' activity (although recurrent ectopic expression cannot be entirely ruled out) and as a result did not use this line for subsequent analyses.

*EGFP* expression in adult fish was most prominent by epifluorescence in cells within the tail, tip of the dorsal fin (Fig. 2f), and peripheral nervous system, particularly observable within the maxillary barbel, a sensory organ, which contains peripheral nerves and neural crest-derived *sox10*-expressing Schwann cells (Fig. 2g). Together, the similar expression patterns of *EGFP* between different stable transgenic lines suggest that our *peak5* transgenic zebrafish reflect endogenous enhancer activity of *peak5* in isolation.

**peak5 is active in early melanoma patches and tumors**. Given that *peak5* is active in NCCs, we next asked if *peak5* is an active enhancer in melanoma. Excitingly, in every transgenic line assessed for tumor growth (Lines A, B, C, and 129B), *EGFP* was expressed, as visualized by epifluorescence on a dissecting scope, in nearly all melanoma tumors (Fig. 3a–f′; Supplementary Table 1) ($n = 67/71$ EGFP positive tumors in 61 animals). Not only is *peak5* active in raised tumors, it is also active in small, unraised, "preclinical" melanoma precursor lesions. Patches of EGFP positive cells, tracked over time, drastically grew similar to *crestin:EGFP* positive patches[7] in most cases (Fig. 3g–j) ($n = 9/15$ EGFP positive patches tracked in 13 animals). While the remaining 6/15 patches did not obviously expand based on gross observation using a dissecting microscope, they remained at least stable and did not clearly regress over a 2–3-month time period observed. EGFP localization in precursor lesions and tumors is particularly striking because *EGFP* is either minimally or not detectably expressed by epifluorescence in adult melanocytes, relative to patches and tumors, in the lines studied. Overall, *EGFP* expression in patches and tumors indicates that *peak5* is an active enhancer in melanoma, and furthermore, represents a *sox10* regulatory element that is active during the early stages of melanoma. To our knowledge, this is the first time a single enhancer region from an evolutionary conserved gene, like *sox10*, has been shown to be active specifically as a transgenic reporter in early melanoma patches and tumors in vivo.

In addition to *peak5*, we generated several stable lines to assess activity of the *sox10* minimal promoter, to serve as a positive control. We also generated stable lines of *peak1* and *peak8*, regions which broadly showed expression in fin mesenchyme and the CNS, respectively, in stable transgenic embryos, to assess

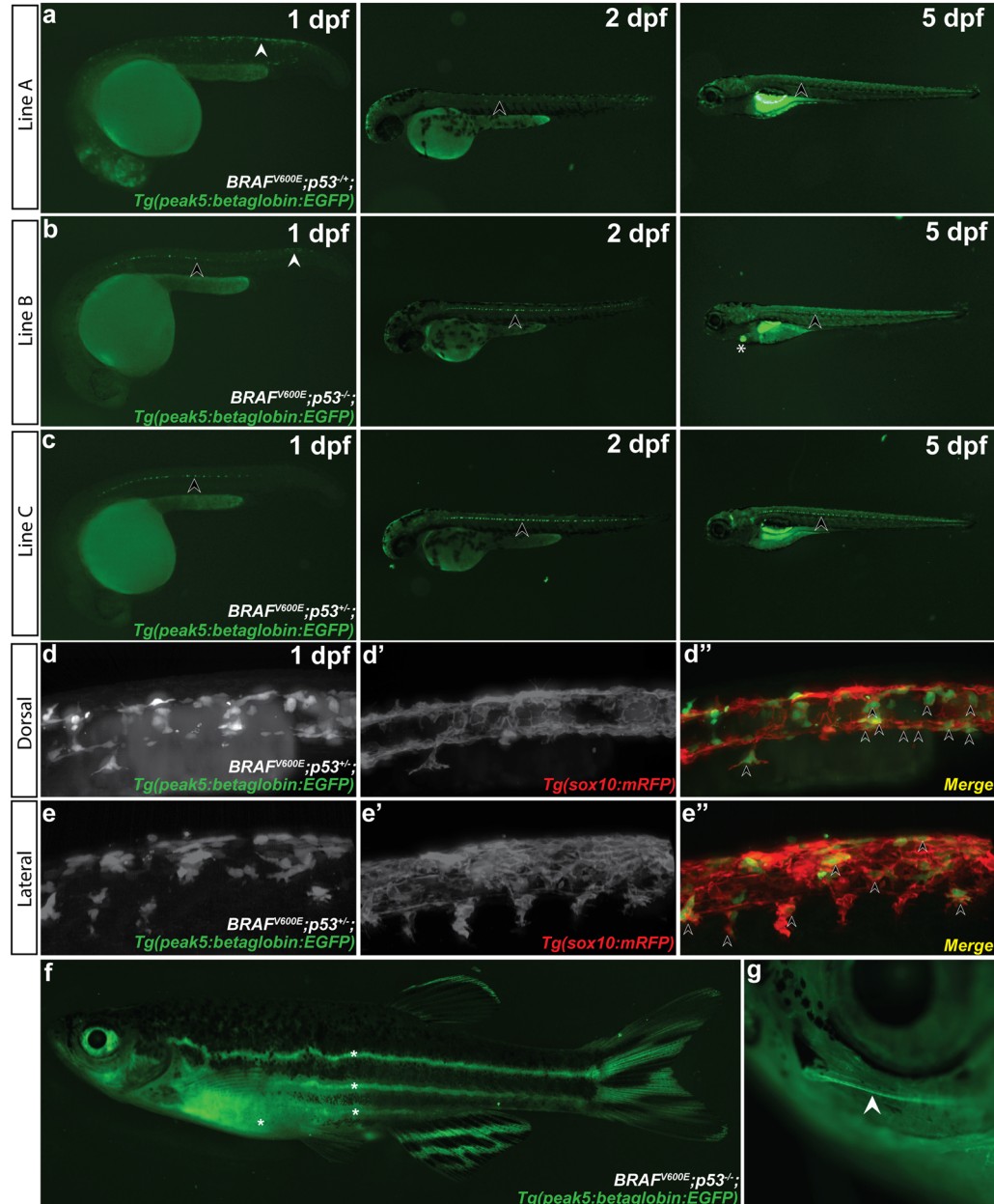

**Fig. 2 *peak5* stable transgenic lines exhibit similar expression patterns in subsets of NCCs and neural crest derivatives.** Multiple transgenic lines derived from independent founder (F0) parental fish were established that transmit the *peak5:betaglobin:EGFP* reporter through the germline. White arrowheads indicate NCCs. Black arrowheads point to KA neurons. Asterisks indicate heart EGFP localization in the heart. **a** Stable line A exhibits strong EGFP localization in NCCs at 1 dpf. At 2 dpf, some NCCs are labeled, and KA neurons are also faintly visible. At 5 dpf, the most prominently labeled cells are KA neurons and cells near or within the swim bladder. **b** At 1 dpf, Line B exhibits EGFP localization in posterior premigratory NCCs and strong localization in KA neurons. By 3 and 5 dpf, EGFP localization is primarily within KA neurons and the heart. **c** Line C mainly exhibits EGFP localization in KA neurons at 1, 2, and 3 dpf. **d** Lightsheet microscopy of the dorsal aspect of *peak5:betaglobin:EGFP* Line A embryos shows *peak5* is active in cells with NCC morphology and premigratory NCC localization at 1 dpf and **d'–d"** co-labels with *Tg(sox10(7.2):mRFP)* positive cells. Arrowheads point to representative co-labeled cells. The lateral aspect also shows **e–e"** EGFP positive cells that co-localize with mRFP positive cells. Arrowheads point to representative co-labeled cells. **f** Whole animal image of *peak5* active in stable transgenic line A. Asterisks (*) indicate regions of auto-fluorescence; a long pass filter set allows discrimination between the yellow hue of autofluorescent tissue and *EGFP*-expressing cells. **g** EGFP positive cells peripheral nervous system cells within the barbel (arrowhead).

activity in melanoma. As predicted, the *sox10* minimal promoter drove *EGFP* expression in melanoma patches and tumors across 6 stable lines ($n = 38/46$ EGFP positive tumors in 42 animals) (Supplementary Fig. 5a-c'; Supplementary Table 1). Unlike *peak5* transgenic lines, *peak1* and *peak8* stable lines expressed *EGFP* embryonically but did not express *EGFP* in nearly all melanoma tumors ($n = 0/6$ EGFP positive tumors in 6 *peak1* transgenic animals from 1 stable line; $n = 1/26$ EGFP positive tumors in 25 *peak8* transgenic animals from three independent stable lines) (Supplementary Fig. 5d-g'; Table 1). These *peak1* and *peak8* data serve as negative controls to highlight the specificity of *peak5* from within the *sox10* super-enhancer and demonstrate that not all sub-elements that drive *EGFP* embryonically are sufficient to drive *EGFP* expression in melanoma. These data also show that

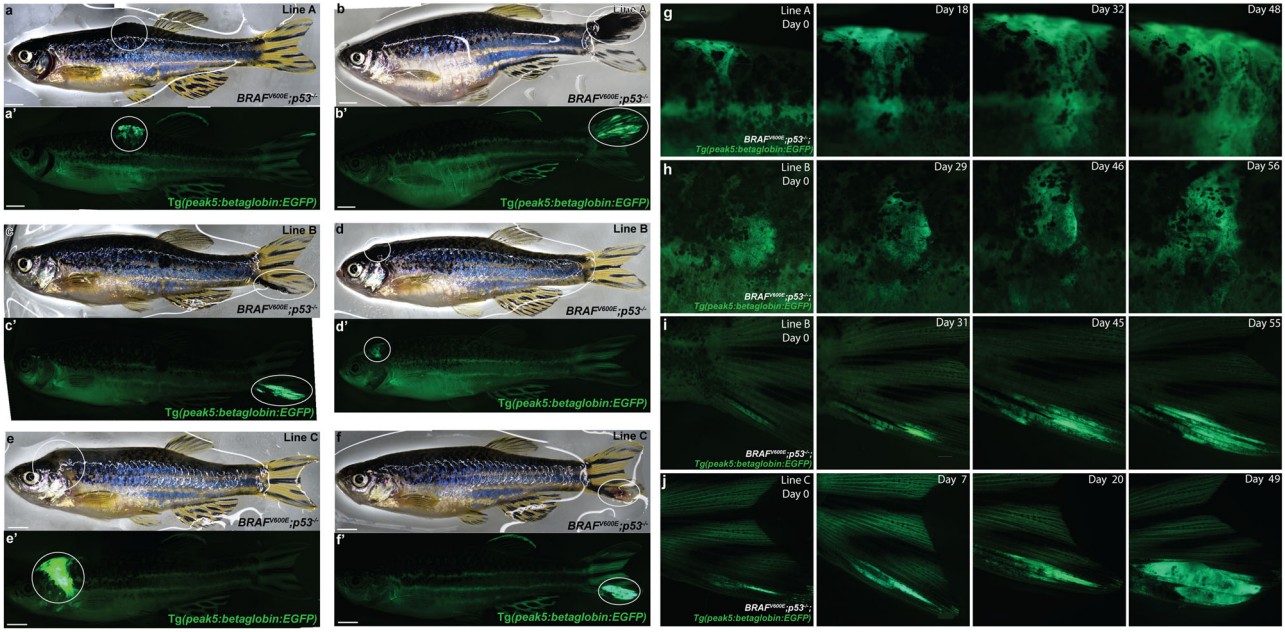

**Fig. 3 *peak5* is active in melanoma tumors across multiple stable lines and in melanoma precursor lesions. a, b** Bright-field images of tumors, circled, in *Tg(peak5:betaglobin:EGFP)* stable line A in the *Tg(BRAF^V600E);p53^−/−* zebrafish background. **a′, b′** Tumors are EGFP positive and *EGFP* is highly expressed in tumors compared to any low *EGFP* expression elsewhere in the fish. **c, d** Bright-field images of *Tg(peak5:betaglobin:EGFP)* stable line B with tumors, circled. **c′, d′** Tumors are EGFP positive. **e, f** Bright-field images of *Tg(peak5:betaglobin:EGFP)* stable line C with tumors, circled. **e′, f′** Tumors are EGFP positive. **g** *Tg (peak5:betaglobin:EGFP)* stable line A zebrafish with dorsally located melanoma precursor lesion that grows over time. **h** *Tg(peak5:betaglobin:EGFP)* stable line B zebrafish with a melanoma precursor lesion in a scale that expands in size over time. **i** *Tg(peak5:betaglobin:EGFP)* stable line B zebrafish with precursor lesion located on the tail that grows over time. **j** *Tg(peak5:betaglobin:EGFP)* stable line C zebrafish with a precursor lesion on the tail that grows into a tumor. Day 0 denotes the first day the precursor lesion was observed.

*peak5* is an enhancer upstream of the *sox10* minimal promoter that is specifically active in melanoma, both in precursor lesions and tumors.

**A conserved sequence within *peak5* is necessary for activity in NCCs and melanoma.** We next asked if within *peak5* there is a smaller sequence that is necessary for activity in neural crest and melanoma. We hypothesized that any nucleotide conservation within *peak5* between zebrafish and other animals would identify a critical regulatory target region. Nucleotide sequence conservation of enhancers between zebrafish and mammals is rare[31]; accordingly, we did not observe any detectable co-linear sequence conservation of *peak5* in humans or mice. However, past sequence comparison of the zebrafish exomes to the exomic sequence of a close relative, the common carp (*Cyprinus carpio*), revealed high conservation of synteny and homologous coding sequences[32]. Thus, we reasoned that evolutionarily important noncoding sequences may also be conserved. Aligning the nucleotide sequence around the *sox10* locus in both zebrafish and carp using Nucleotide BLAST revealed discrete regions of conservation, including *peaks 2, 4, 5, 8*, the minimal promoter, a small portion of *peak13*, as well as most of the *sox10* coding region (Fig. 4a). Interestingly, these conserved regions correspond to the enhancers with the most robust *EGFP* expression in F0 transgenic embryos (Supplementary Fig. 2).

Within *peak5*, we identified a 192 bp region that is conserved between carp and zebrafish, as well as other selected members of the *Cyprinidae* family (*Sinocyclocheilus grahami*, *Pimephales promelas*, *Carassius auratus*, and *Oxygymnocypris stewartii*) (Fig. 4b). Regions of conservation of *peak5* between species other than carp were located on whole-genome shotgun contigs through Nucleotide BLAST. Therefore, to verify that these regions of conservation are indeed near *sox10* in these other

species, we aligned each identified scaffold to the zebrafish *sox10* locus. We identified proximity to the *sox10* locus and/or synteny with other putative *sox10* regulatory regions for all other examined species except *Pimephales promelas* (Supplementary Fig. 6).

To test if this *peak5* conserved sequence is required for *peak5* activity, we deleted the sequence within the reporter assay plasmid containing the wild-type *peak5* sequence (Fig. 4c). Interestingly, compared to full-length wild-type *peak5* sequence, which exhibits activity in NCCs and in KA neurons embryonically, embryos injected with the 192 bp conserved sequence deletion construct (*peak5Δ192:betaglobin:EGFP*) exhibited diminished *peak5* activity in NCCs, while maintaining strong EGFP localization in KA neurons (Fig. 4d–h′). This suggests that this core conserved sequence controls *peak5* activity at least within a subset of NCCs. Injection of only the *peak5* conserved sequence (*peak5_conserved:betaglobin:EGFP*) resulted in labeling of only NCCs, albeit at weaker levels compared to full-length *peak5*, demonstrating that the conserved sequence is sufficient for *peak5* activity in NCCs and is not sufficient for labeling of KA neurons (Fig. 4d, i, j′).

In accordance with these mosaic analyses, embryos from *peak5Δ192* stable transgenic lines exhibit EGFP localization in KA neurons, but no clear localization in NCCs located in the dorsal posterior region of the trunk (Fig. 4k–k′; Supplementary Fig. 7a-b). In contrast, NCCs in *peak5_conserved* stable transgenic embryos are EGFP positive and KA neurons are not labeled (Fig. 4l–l′ Supplementary Fig. 7c-d). We did not observe EGFP positive barbels in *peak5Δ192* stable transgenic adult animals (Fig. 4m; Supplementary Fig. 8c); however, in adult stable transgenic *peak5_conserved* zebrafish, cells within the maxillary barbel are labeled, similar to wild-type *peak5* stable lines (Fig. 4n; Supplementary Fig. 8a-b). Furthermore, the majority of tumors in

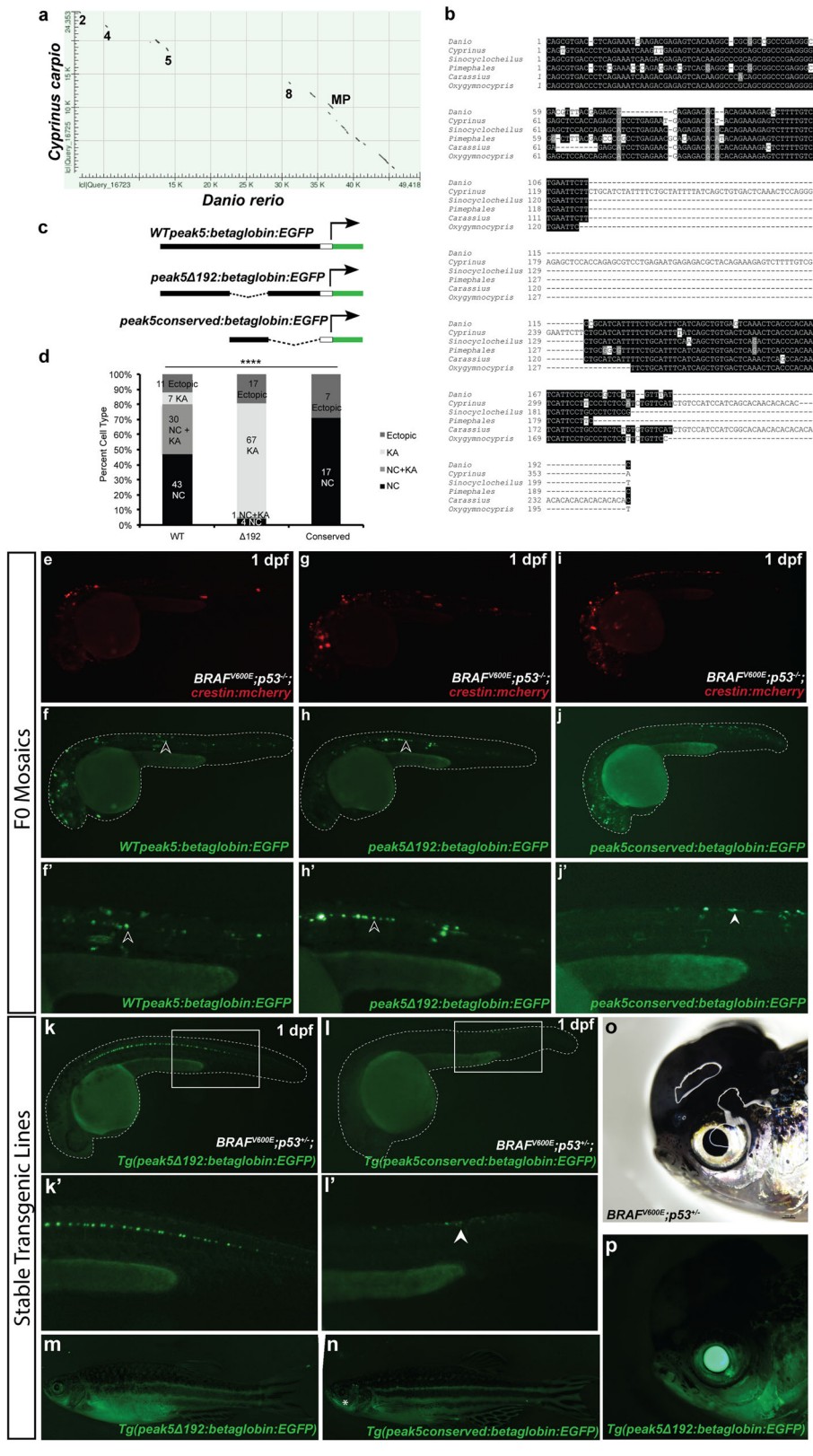

adult *peak5Δ192* stable transgenic animals are not EGFP positive (*n* = 33/36 EGFP negative tumors in 32 animals from 3 different stable transgenic lines) (Fig. 4o, p; Supplementary Table 1). These data suggest that the conserved sequence we identified is not only necessary for *peak5* activity in a subset of NCCs, but also is necessary for robust *peak5* enhancer activity in melanoma.

**SoxE TFBS regulate *peak5* activity in NCCs.** We next asked which specific transcription factor binding sites (TFBS) within this conserved region of *peak5* are functionally necessary for activity. Out of a list of predicted TFBS, some of which have roles in NCC development, our attention was drawn to predicted dimeric SoxE binding sites (Fig. 5a), as it was previously noted that such dimeric

**Fig. 4 A conserved region of *peak5* is necessary for neural crest and melanoma activity. a** Dot-matrix view of the alignment of the carp (*Cyprinus carpio*) *sox10* coding and surrounding noncoding genomic region of *sox10* compared to the same genomic region in zebrafish (*Danio rerio*). Numbers indicate regions where peak sequences are conserved. MP minimal promoter. **b** Sequence alignment of the most conserved region of *peak5* to members of the *Cyprinidae* family. **c** Schematic of plasmids used in the experiment. **d** Quantification of the percentage of categorizable EGFP + embryos exhibiting NC, KA, or both NC and KA labeling, or ectopic expression in screened *WT peak5* (*n* = 91), *peak5Δ192* (*n* = 89), and *peak5_conserved* (*n* = 24) injected embryos. NC neural crest. KA Kolmer–Agduhr neurons. ****\*\*\*\*p*-value <0.0001, Chi-squared analysis. Injection of *crestin:mCherry* simply served as a positive injection control in this experiment. **e** *crestin:mch* expression in an embryo injected with *WT peak5* and *crestin:mch*. **f** *WT peak5* is active in both neural crest and **f'** KA neurons. **g** *crestin:mch* expression in an embryo injected with *peak5Δ192* and *crestin:mch*. **h** *peak5Δ192* is not active in NCCs, but **h'** is active in KA neurons. **i** *crestin:mch* expression in an embryo injected with the conserved *peak5* sequence and *crestin:mch*. **j** The conserved *peak5* sequence is active in neural crest and **j'** does not exhibit activity in KA neurons. Black arrowheads indicate KA neurons and white arrowheads indicate NCCs. **k–k'** *Tg(peak5Δ192:EGFP)* embryos exhibit EGFP localization in KA neurons, but no overt localization in NCCs in the trunk is observed. **l–l'** Contrastingly, *Tg(peak_conserved:EGFP)* embryos do not have labeled KA neurons, but exhibit EGFP positive cells in the trunk of the embryo (white arrowhead). **m** Little EGFP localization is observed in *Tg(peak5Δ192:EGFP)* adult fish whereas **n** cells within the maxillary barbels (asterisk) of *Tg(peak_conserved:EGFP)* are EGFP positive. **o, p** The majority of tumors that develop in *Tg(peak5Δ192:EGFP)* are EGFP negative.

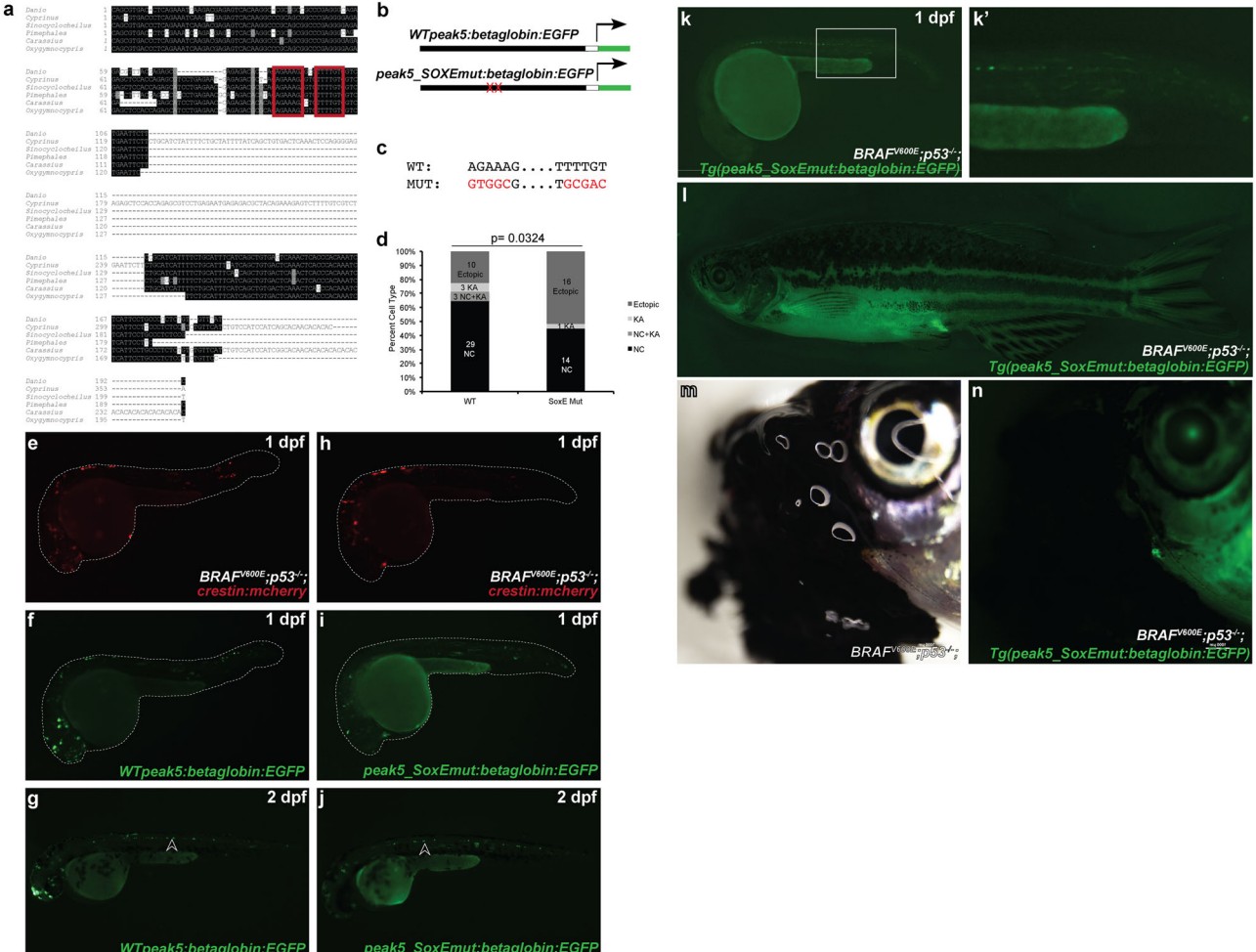

**Fig. 5 Mutation of dimeric SoxE TFBS affect *peak5* neural crest activity. a** Dimeric SoxE TFBS, as predicted by JASPAR (outlined in red), are present in the conserved sequence of *peak5*. **b** Schematic of plasmids used in the experiment. Mutations to SoxE TFBS were made in the context of the full-length WT *peak5* plasmid. **c** Five out of six nucleotides for each motif were mutated. **d** Quantification of the percentage of categorizable EGFP + embryos exhibiting NC, KA, both NC, and KA, or ectopic expression in screened *WT peak5* (*n* = 45) and *peak5_SoxEmut* (*n* = 31) injected embryos. NC neural crest, KA Kolmer–Agduhr neurons. *p*-value = 0.0324, Fisher's exact test. Injection of *crestin:mCherry* simply served as a positive injection control in this experiment. **e** *crestin:mch* expression in an embryo injected with *WT peak 5* and *crestin:mch*. **f** *WT peak5* is strongly active in neural crest in F0 mosaics at 1 dpf. **g** *WT peak5* is strongly active in neural crest-derived cells and KA neurons at 2 dpf. **h** *crestin:mch* expression in an embryo injected with *peak5_SoxEmut* and *crestin: mch*. **i** *peak5_SoxEmut* is not active in the neural crest in F0 mosaics, but **j** is active in KA neurons. Arrowheads indicate KA neurons. **k–k'** Stable transgenic *peak5_SoxEmut* lines exhibit *peak5* activity in KA neurons, but do not exhibit labeling in posterior premigratory neural crest cells. **l** Adult stable transgenic lines do not exhibit *peak5* activity in barbel peripheral nerves. **m, n** Tumors that develop in *Tg(peak5_SoxEmut:EGFP)* lines are EGFP negative.

SoxE binding sites exhibit functional relevance in mouse *Sox10* enhancers[33]. Furthermore, dimeric SOXE TFBS with 3–5 bp spacers are enriched in human melanoma cell line-associated putative enhancers identified by DNase hypersensitivity site mapping[34]. We therefore mutated both of these predicted SoxE binding sites within the *peak5* wild-type sequence to functionally test their necessity (Fig. 5b, c). Mutated plasmid was co-injected with *crestin:mCherry*, to serve as a positive injection control, and scored for activity in NCCs and KA neurons. Compared to *peak5* wild-type injected embryos, mutation of the SoxE binding sites results in drastically diminished *EGFP* expression in NCCs, but KA *EGFP* expression is maintained (Fig. 5d–j). Generation of two independent stable transgenic lines, dubbed *Tg(peak5_SoxEmut:betaglobin:EGFP)*, also demonstrated that the dimeric SoxE binding sites are necessary for *peak5* activity in premigratory NCCs. Embryos exhibit *peak5*-driven enhancer activity in KA neurons, but no detectable activity is observed in posterior premigratory NCCs (Fig. 5k–k'). Low-level ectopic activity in muscle was present in one of two lines (Fig. 5k; Supplementary Fig. 7e). In adults, while wild-type *peak5* activity is robust in *sox10*-expressing peripheral nerves within barbels, *EGFP* is minimally expressed in *Tg(peak5_SoxEmut:betaglobin:EGFP)* barbels (Fig. 5l; Supplementary Fig. 8d), further illustrating that mutation of dimeric SoxE TFBS significantly diminishes *peak5* activity. In addition, EGFP is not visualized under the same microscopy conditions in the majority of tumors from these two independent stable lines ($n = 11/13$ EGFP negative tumors in 11 animals), further highlighting that the SoxE dimeric binding site is necessary for robust *peak5* activity in melanoma (Fig. 5m, n; Supplementary Table 1). This mutational analysis confirms that the dimeric SoxE binding sites are functional and likely important for the transcriptional regulatory activity of *peak5* in the neural crest and in melanoma.

**In vivo deletion of *peak5* decreases *sox10* expression and alters adult stripe patterning.** Finally, to more definitively assess if *peak5* is indeed a functional enhancer regulating endogenous *sox10* gene expression, we designed single-guide RNAs (sgRNAs) to use CRISPR/Cas9 to delete the endogenous *peak5* genomic sequence. As a positive control, we also designed sgRNAs to delete the *sox10* minimal promoter (*sox10min*), containing the transcriptional start site, based upon our ATAC-seq data. We injected sgRNAs flanking either *sox10min* or *peak5* and Cas9 protein into *Tg(BRAF^{V600E};crestin:EGFP);p53^{-/-}* or *Tg (BRAF^{V600E});p53^{-/-}* zebrafish embryos. Efficacy of the CRISPR was assessed by PCR analysis of the targeted genomic region at 1 dpf, demonstrating mosaically deleted targeted loci as evidenced by the laddering of DNA in the CRISPR injected embryos (Fig. 6a, b). To identify carriers of stable *sox10min* or *peak5* deletion alleles, F0 injected embryos were raised to maturity and outcrossed to wild-type *Tg(BRAF^{V600E});p53^{-/-}* zebrafish. F1 embryos were then screened for deletions through PCR and Sanger sequencing analyses. We identified one *sox10min* 200 bp deletion allele, *stl792*, and three *peak5* deletion alleles (Fig. 6c, d; Supplementary Fig. 9a). Our previous enhancer analyses of *peak5* demonstrated that the conserved *peak5* sequence is necessary for neural crest and melanoma activity of *peak5*, therefore we chose to focus our analysis on the *stl538* allele which primarily deletes the conserved sequence of *peak5*.

To determine if deletion of the *sox10* minimal promoter and *peak5* affects *sox10* expression, we utilized whole-mount in situ hybridization (WISH) for *sox10* mRNA at 1 dpf to visualize *sox10* expression distribution and levels. As predicted, control *sox10min* 1 dpf mutant embryos revealed strikingly complete loss of *sox10* expression compared to wild-type and heterozygous controls (Fig. 6e, f) and phenocopies the *colourless* mutant[35] (Supplementary

Fig. 9f-g). Embryos were scored as either "strong," "reduced," or "absent" *sox10* expression in WISH. One hundred percent of *sox10min* homozygous mutants did not express any *sox10* as compared to wild-type and heterozygous siblings (Fig. 6f, g). Interestingly, the well-studied *colourless* mutant in the *sox10* gene shows a reduction in *sox10* expression in the homozygous state by WISH and has no apparent heterozygous phenotype[35], compared to our *sox10min* promoter deletion mutation which leads to decreased *sox10* expression and a phenotype in the heterozygous state, as well as complete loss of *sox10* expression by WISH in homozygous mutants (see below). From this we conclude that our ATAC-seq data can identify functionally important noncoding genomic regions that control *sox10* expression.

Previous analyses simultaneously targeting three *sox10* enhancers in zebrafish using CRISPRi, not including *peak5*, showed slight decreases in *sox10* expression in F0 embryos[23]. We therefore predicted that deletion of *peak5* would result in a more subtle decrease in *sox10* expression because of the multiple putative enhancers associated with *sox10*. Indeed, we observed a noticeable decrease in global *sox10* expression in 1 dpf WISH for *sox10* in *peak5* mutant embryos compared to wild-type and heterozygous controls (increased "reduced/slightly reduced" percentage of scored embryos in homozygous mutant compared to heterozygous or wild-type controls) (Fig. 6h–j). This decrease in *sox10* expression supports our hypothesis that *peak5* is a bona fide functional enhancer for the endogenous *sox10* gene.

Given this embryonic phenotype showing reduced *sox10* expression, we raised *peak5* (*stl538* deletion allele) wild-type, heterozygous, and mutant animals to adulthood to observe if this decrease in *sox10* expression results in a neural crest/pigment cell phenotype in adults. We studied the *peak5* deletion allele in the *Tg(BRAF^{V600E});p53^{-/-}* background, and found that wild-type *peak5* zebrafish exhibit few breaks in the horizontal stripe pattern (median 1 break per fish, interquartile range (IQR) 0–2 breaks) compared to *peak5* homozygous deletion mutant sibling adult fish (*stl538/stl538*) which exhibited a drastically increased number of stripe breaks (median 4 breaks per fish, IQR 3–8 breaks) (Fig. 6k–m). Such a pigment patterning defect is a hallmark of dysregulated neural crest/pigment cell development such as seen in Waardenburg syndrome[36,37], which we interpret to be consistent with reduction in a key regulator of this process, *sox10*, and together with the embryonic *sox10* expression data, support the conclusion that *peak5* is a *bona fide* enhancer regulator *sox10* expression.

Lastly, to test further if *peak5* and the *sox10* minimal promoter cooperate together genetically to influence *sox10* activity, we generated *Tg(BRAF^{V600E});p53^{-/-};peak5^{stl538/+};sox10min^{stl792/+}* trans-heterozygotes. Compared to *Tg(BRAF^{V600E});p53^{-/-}* wild-type siblings (Fig. 6n), *peak5^{stl538}* heterozygotes (Fig. 6o), and *sox10min^{stl792}* heterozygotes (Fig. 6p), *peak5^{stl538/+};sox10-min^{stl792/+}* trans-heterozygotes exhibit a drastic increase in the number of stripe breaks, predominantly anteriorly (Fig. 6q). *Tg (BRAF^{V600E});p53^{-/-}* wild-type fish and *peak5^{stl538}* heterozygous fish exhibit few stripe breaks in the horizontal stripe pattern (median 0 breaks per wild-type fish, IQR 0–3.25; median 3 stripe breaks per *stl538* heterozygous fish, IQR 1.5–4). *sox10min^{stl792}* heterozygotes exhibit a statistically significant increase in the number of stripe breaks compared to wild-type fish (median 5.5 stripe breaks per fish, IQR 3.75–8.25). The *sox10min^{stl792/+}* heterozygote phenotype is notable because other heterozygous *sox10* alleles (*colourless*) in zebrafish do not exhibit a pigmenta-tion phenotype[35]. An even more drastic increase in the number of stripe breaks was observed in *peak5^{stl538/+};sox10min^{stl792/+}* trans-heterozygotes (median 9.5 stripe breaks per fish, IQR 7.75–12.75) (Fig. 6r). Together, these data show that trans-heterozygotes of regulatory elements of *sox10*, *peak5* and *sox10min*, interact

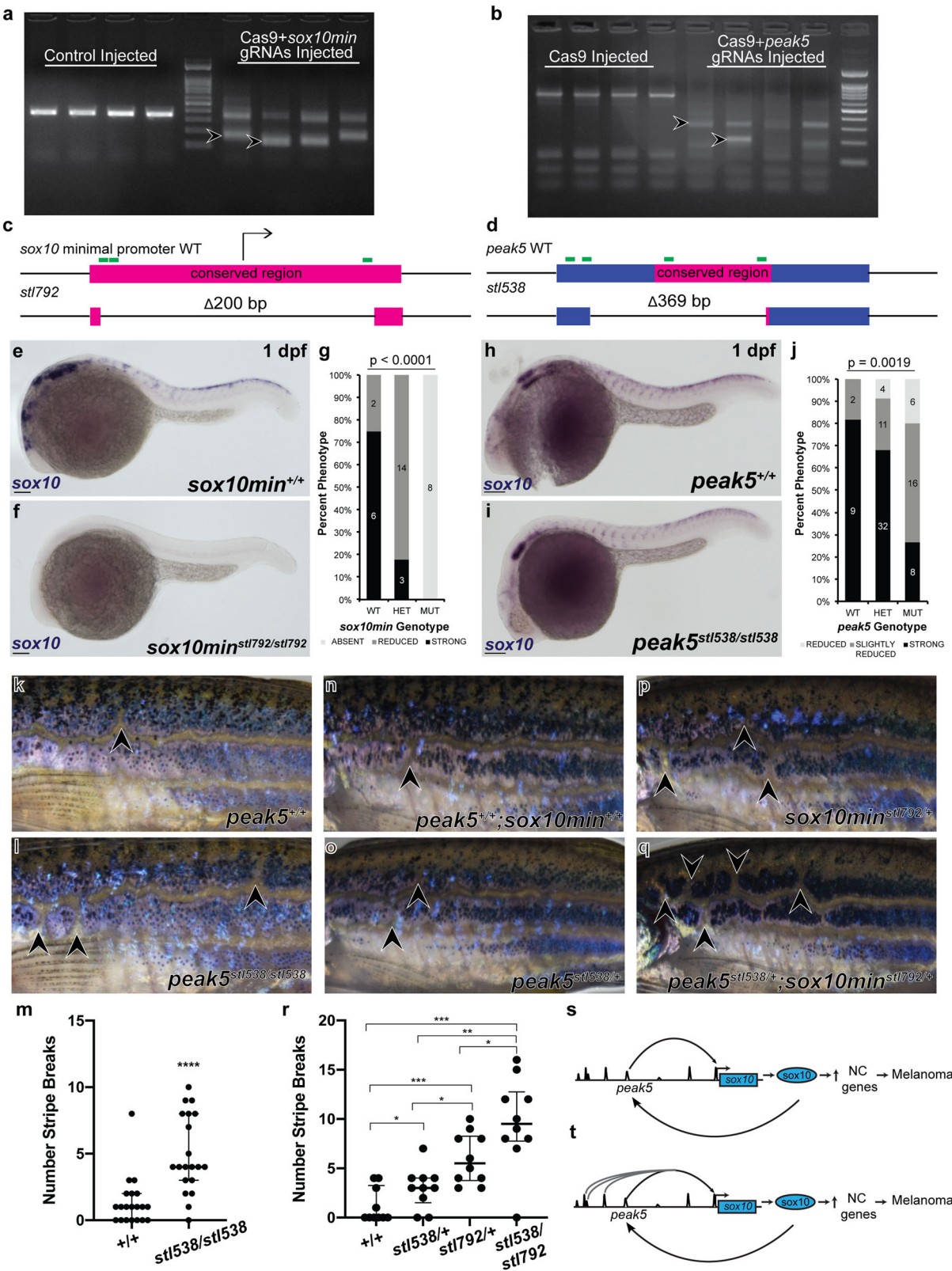

genetically to produce an additive phenotypic effect on pigment cell patterning.

## Discussion

Melanoma initiating cells express aspects of a neural crest program that is maintained as a tumor progresses[7]. One gene within the neural crest program, *sox10*, is a critical regulator of NCC development and is also upregulated in both zebrafish and human melanomas. Indeed, human melanomas immunostain positive for SOX10 in most cases, even more often than the key melanocyte lineage regulator MITF[38–40]. Past work demonstrated that modulation of *sox10* expression in melanocytes, which normally

**Fig. 6 In vivo CRISPR/Cas9 deletion of *peak5* affects *sox10* expression and adult stripe patterning. a** 1 dpf CRISPR/Cas9 injected embryos with sgRNAs targeting the *sox10* minimal promoter show DNA laddering (arrowheads) as compared to control injected embryos. **b** CRISPR/Cas9 injected embryos with sgRNAs targeting *peak5* show complete disruption of the wild-type locus as evidenced by DNA laddering (arrowheads) compared to Cas9 injected embryo. **c** The *stl792* allele contains a 200 bp deletion at the *sox10* minimal promoter locus, which contains the TSS of *sox10*. **d** The *stl538* allele contains 369 bp deletion at the *peak5* locus, mainly disrupting the conserved region of *peak5*. **e** WISH for *sox10* mRNA shows robust expression in wild-type 1 dpf embryos and **f** absent expression in *stl729* homozygous mutant 1 dpf embryos. **g** Quantification of WISH expressing the percent of embryos displaying either "strong," "reduced," or "absent" *sox10* expression. $p < 0.0001$, Fisher's exact test. **h** WISH for *sox10* mRNA shows robust expression in wild-type 1 dpf embryos and **i** reduced expression in *stl538* homozygous mutant 1 dpf embryos. **j** Quantification of WISH expressing the percent of embryos displaying either "strong," "reduced," or "strongly reduced" *sox10* expression. $p = 0.0019$, Fisher's exact test. **k** Lateral view of one side of a wild-type $Tg(BRAF^{V600E});p53^{-/-}$ adult fish with 1 stripe break (arrowhead) compared to **l** a *stl538* homozygous adult fish with 3 stripe breaks (arrowheads). **m** Quantification of the number of stripe breaks per fish. Each dot represents one fish. Wild-type fish exhibit a median of 1 stripe break per fish, IQR 0–2. *stl538* homozygous fish exhibit a median of 4 stripe breaks per fish, IQR 3–8. *****$p < 0.0001$, Welch's *t*-test. **n** Lateral view of one side of a wild-type $Tg(BRAF^{V600E});p53^{-/-}$ adult fish with 1 stripe break (arrowhead) compared to **o** a *peak5*$^{stl538}$ heterozygote with two stripe breaks (arrowheads), **p** a *sox10min*$^{stl792}$ heterozygote with 3 stripe breaks (arrowheads) and **q** a *peak5*$^{stl538}$;*sox10min*$^{stl792}$ trans-heterozygote with 5 stripe breaks (arrowheads). **r** Quantification of the number of stripe breaks per fish. Each dot represents one fish. Wild-type fish exhibit a median of 0 stripe breaks, IQR 0–3.25. *peak5*$^{stl538}$ heterozygotes exhibit a median of 3 stripe breaks, IQR 1.5–4. *sox10min*$^{stl792}$ heterozygotes exhibit a median of 5.5 stripe breaks per fish, IQR 3.75–8.25. *peak5*$^{stl538/+}$;*sox10min*$^{stl792/+}$ trans-heterozygotes exhibit a median of 9.5 stripe breaks per fish, IQR 7.75–12.75. *$p < 0.05$, **$p = 0.001$, ***$p < 0.0002$, Welch's *t*-test. **s** In our model, *peak5* activates *sox10* expression in NCCs during development and then again in melanoma precursor cells, likely through dimeric SoxE TFBS. This leads to upregulation of neural crest-related genes and melanoma onset. **t** *peak5* is most likely not the only *sox10* enhancer that regulates *sox10* expression. Multiple elements within the *sox10* super-enhancer may coordinate together to upregulate *sox10* expression during melanoma initiation.

express *sox10* at low levels, affects melanoma onset rates[7,21,38], illustrating that regulation of *sox10* plays a key role during melanoma initiation. Therefore, deciphering the transcriptional regulation of *sox10* may reveal earlier mechanisms of melanoma initiation that drive NCP activation. Ongoing studies continue to decipher the embryonic function of *sox10* in zebrafish[23,35,41], but to date, regulation of *sox10* expression in melanoma is poorly understood. In this study, we screened individual putative *sox10* regulatory elements for activity within NCCs and melanoma in zebrafish. From this analysis, we identified *peak5*, an enhancer that is active in embryonic neural crest and melanoma, including in precursor lesions.

We used ATAC-Seq analysis of multiple zebrafish melanoma tumors to identify recurrently open chromatin domains surrounding and within the *sox10* locus as putative enhancers. The chromatin landscape surrounding *sox10* closely mirrors the previously reported ATAC-Seq profile of zebrafish NCCs (Supplementary Fig. 1B) by the 5-6 somite stage, although is not evident at earlier stages (e.g., bud or 75% epiboly) suggesting that the accessibility landscape near *sox10* seen in the melanoma tumors may be most reflective of one established in late neural crest progenitors committed to a pigment cell progenitor lineage[23–26]. Injection of reporter constructs harboring these putative enhancers into embryos revealed that 9 out of 11 displayed some level of activity within NCCs, as compared to injected negative controls, indicating their potential as active enhancers in melanoma. Generation of multiple stable lines revealed that *peak5*, a 669 bp sequence from 22.8 kb upstream of the *sox10* transcriptional start site, is active embryonically in a subset of NCCs and KA neurons. In transgenic zebrafish adults, *peak5* is at most minimally active in melanocytes in vivo, where transgenic *EGFP* expression is not evident, in the context of a transgenic EGFP reporter assay but is highly active and readily visualized in melanoma patches and tumors. Furthermore, we identified a region within *peak5* that is conserved between members of the *Cyprinidae* family and mediates the NCC activity of *peak5*. Within this conserved region, dimeric SoxE binding sites are critically necessary for *peak5* activity within the neural crest and melanoma. Lastly, we definitively demonstrate that *peak5* is a functional enhancer of *sox10* through in vivo CRISPR deletion of *peak5*, and to our knowledge is one of few examples of a CRISPR engineered genomic deletion of a functional transcriptional control element in zebrafish[42–44]. Deletion of the conserved region of *peak5* results in global

decreases in *sox10* expression at 1 dpf in a $Tg(BRAF^{V600E});p53^{-/-}$ background as shown through WISH, and homozygous *peak5* mutant adults also display a pigment/stripe patterning defect. A similar and even more pronounced stripe patterning defect is observed in fish heterozygous for a *sox10* minimal promoter deletion. The stripe phenotype is further exacerbated in *peak5* and *sox10min* trans-heterozygotes, demonstrating that *peak5* and the *sox10* minimal promoter genetically interact. Together, *peak5* serves as a proxy enhancer to understand how transcriptional regulation of *sox10* and potentially other melanoma enriched genes may influence melanoma initiation.

The identification of *crestin* as an early marker of melanoma, whose expression is highly dependent on intact *sox10* transcription factor binding sites, provided a live marker of the earliest detectable events of melanoma initiation and confirmed that the NCP activation is an early step in the process[7]. This study expands upon this knowledge by newly identifying a regulatory element that can also be utilized as a specific in vivo specific reporter of melanoma formation. Importantly, unlike *crestin*, *sox10* is conserved across all vertebrate species; therefore, identifying *sox10* enhancers that are active in melanoma in zebrafish could more readily lead to the identification of human *SOX10* enhancers that influence melanoma onset. While we were unable to identify stretches of co-linear conserved sequence between the zebrafish and human/mouse upstream regions of *SOX10*, predicting the regulatory function of enhancers based on sequence conservation alone is often not possible, such as in the case of *RET* enhancer elements, which function faithfully between human and zebrafish in reporter assays, but are not conserved in an obvious way at the primary nucleotide level[45].

Further, we were intrigued to find that while zebrafish *sox10* enhancers are not conserved in a co-linear sequence with potential human *SOX10* enhancers, several zebrafish enhancers are conserved by nucleotide sequence between members of the *Cyprinidae* family. Conservation of *peaks 2, 4, 5, 8, 13*, and the *sox10* minimal promoter between zebrafish and carp suggest an evolutionary pressure to maintain these regulatory sequences. Interestingly, the highest region of conservation for each peak is oriented near the center of the ATAC-Seq peak, including *peak5*.

This region of high conservation within *peak5* also leads to the hypothesis that the key TFBS controlling *peak5* activity in a subset of NCCs and melanoma are also conserved. Identification of SoxE dimeric binding sites points toward potential

transcription factors, most notably Sox10, that may bind and activate *peak5*. Similar to deletion of the conserved sequence within the context of full-length *peak5*, mutation of both SoxE TFBS diminishes *peak5* activity in posterior premigratory NCCs but does not affect *peak5* activity within KA neurons. The dimeric SoxE mutations also affect *peak5*-driven reporter activity in melanoma. Most tumors we observed in two independent stable transgenic lines were EGFP negative, indicating these two TFBS play a significant role, even in the context of a ~600 bp regulatory element, in modulating *peak5* activity in melanoma. Therefore, members of the SoxE family (most likely Sox10 or Sox9 which are expressed in the melanocyte lineage) are prime candidate transcription factors that may bind and activate *peak5*. The most stringent test of this prediction would require single-cell level ChIP-Seq analysis for these factors on in vivo-isolated tumor-initiating cells, a technical feat not feasible for this tissue at present. Dimeric SOXE binding sites are overrepresented in human melanoma cells in regions of open chromatin[34] and have also been observed near promoters of genes regulated by Sox10[46], suggesting that Sox10 may bind the dimeric sites in *peak5*. Mutation of the dimerization domain within Sox10 in mice also causes defects in melanocyte development, underscoring the importance of dimeric Sox10 binding[47].

Interestingly, dimeric SoxE sites have also been shown to be necessary for activity of two mouse *Sox10* enhancers, MCS4 and MCS7[33]. Since zebrafish *sox10* enhancers are not conserved by sequence with mouse *Sox10* enhancers, it will be interesting to test if either mouse MCS4 or MCS7 *Sox10* enhancers have overlapping embryonic and adult activity patterns with *peak5* in zebrafish. Moreover, examining the other functional TFBS present in *peak5* in zebrafish may help predict and identify a *peak5*-equivalent enhancer in humans that is dependent on the same or similar complement of TFBS.

One caveat of determining the functionality of a putative enhancer is the potential to identify false positives. In zebrafish, random genome integration of enhancers using the Tol2 system can result in enhancer trapping[29]. Therefore, in addition to analyzing each putative *sox10* enhancer in numerous individual F0 transgenic embryos, thus sampling an array of insertion sites, we also generated multiple stable lines for *peak5*, which exhibited highly similar expression patterns, supporting that our lines reflect bone fide spatial and temporal activity of *peak5* embryonically and in melanoma. Variations amongst transgene integration site in different stable lines, leading to subtle position effects in addition to genomic instability in melanoma tumors, likely also explain why we infrequently observe EGFP negative tumors in stable lines that predominately express EGFP in tumors (*peak5* n = 4/71 EGFP negative tumors) and a small number of EGFP positive late-stage tumors in *peak8* (n = 1/26) and *peak5Δ192* (n = 3/36) stable lines. Similarly, differences in expression levels of *EGFP* in all transgenic animals may be due to differences in the copy number of the transgene.

To truly test if *peak5* is a functional enhancer, we used CRISPR to delete *peak5* in vivo. The resulting decreased embryonic *sox10* expression and adult stripe patterning defect demonstrate that *peak5* represents a bone fide enhancer of *sox10*. Interestingly, we also observed that newly generated null allele of *sox10*, *stl792*, which deletes the 350 bp minimal promoter of *sox10*, exhibits a heterozygous stripe phenotype, which is more penetrant than the lack of phenotype previously described in heterozygous zebrafish *sox10* alleles (*colourless*)[35]. The *peak5* and *sox10* minimal promoter deletions create a foundation to begin to understand how specific enhancer states epigenetically modulate disease states in vivo. Future work will aim to characterize and investigate the mechanism by which *peak5* deletion produces the stripe patterning defect and if this deletion affects melanoma tumor onset

rates. Of note, the new alleles we generated were made in the background of *Tg(BRAF^{V600E});p53^{−/−}*, which generates an aberrant melanocyte state due to hyperactivation of BRAF and the MAPK pathway. We hypothesize that the *Tg(BRAF^{V600E});p53^{−/−}* functions as a sensitized background that permits the effects of *peak5* or *sox10* minimal promoter deletions to emerge. We have outcrossed the *peak5 stl538* allele into a wild-type background and by gross visualization do not see apparent stripe defects. It will be interesting to further understand the effects of these deletions in a true wild-type background as well as understand how cells in a sensitized, or preneoplastic disease state, may be more susceptible to effects of transcriptional regulation of *sox10*.

Finally, we propose a model by which *peak5* contributes to increased *sox10* expression in melanoma initiation and progression (Fig. 6s, t). In this model, the *peak5* region is open and enhances *sox10* expression in early melanoma precursor lesions and, through a feed-forward mechanism, autoregulates increasing *sox10* expression by binding at the dimeric SoxE sites in *peak5*. An appealing aspect of this type of model is that even stochastic perturbations that increase *sox10* binding/activation ability would tend to be reinforced and amplified, potentially "locking in" a *sox10* high/pro-NCC program in melanoma. Again, while stretches of conserved, noncoding DNA are most often not readily identifiable by comparison between humans/mouse and zebrafish, the presence of such dimeric SoxE sites in bona fide murine NCC enhancers[33] and putative human melanoma enhancers[34] is consistent with this being a general mechanism by which Sox10 target genes are upregulated in melanoma. Given that *peak5* is not active in all cells that express *sox10* during embryogenesis, other *sox10* enhancers likely also contribute to *sox10* regulation during melanoma initiation. Together, developmentally incorrect activation of *sox10* enhancers in zebrafish and human melanomas suggests that understanding the activity and contribution of individual enhancers within this locus will illuminate earlier initiating steps of melanoma. This study identifies one of these enhancers, *peak5*, that is active embryonically and in melanoma, and lays the groundwork to further dissect the role of other *sox10* enhancers in zebrafish and higher vertebrates in melanoma initiation.

## Methods

**Zebrafish lines and rearing conditions**. Zebrafish, *Danio rerio*, were reared in accordance with Washington University IACUC animal protocols in the Washington University Zebrafish Consortium Facility. Adult zebrafish were crossed either as pairs or groups, and embryos were raised in egg water (5 mM NaCl, 0.17 mM KCl, 0.33 mM CaCl₂, 0.33 mM MgSO₄) at 28.5 °C. Larvae were staged at days post fertilization (dpf) and adults were staged at months post fertilization (mpf). The following strains and transgenic zebrafish were used in this study: AB*, *Tg(BRAF^{V600E};crestin:EGFP);p53^{−/−}*[7], *Tg(BRAF^{V600E});p53^{−/−}*[6], *Tg(sox10(7.2): mRFP)*[30], *Tg(crestin:mcherry), Tg(peak5:betaglobin:EGFP), Tg(peak5Δ192:betaglobin:EGFP), Tg(peak5_conserved:betaglobin:EGFP), Tg(peak5_SoxEmut:betaglobin: EGFP), Tg(sox10min:EGFP), Tg(peak1:betaglobin:EGFP), Tg(peak8:betaglobin: EGFP), Tg(BRAF^{V600E});p53^{−/−};sox10min^{stl792}, Tg(BRAF^{V600E});p53^{−/−};peak5^{stl536}, Tg(BRAF^{V600E});p53^{−/−};peak5^{stl537}*, and *Tg(BRAF^{V600E});p53^{−/−};peak5^{stl538}*.

**Isolation of bulk tumors**. Following humane euthanasia, grossly visible bulk melanoma tumors from *Tg(BRAF^{V600E}; crestin:EGFP);p53^{−/−}* zebrafish were excised with a razor, manually sheared, and incubated in fresh 0.9X PBS with 2.5 mg/mL Liberase for up to 30 min to dissociate cells. Fetal bovine serum addition terminated the digestion, and the cells were passed through a 40-mm filter. Cells were centrifuged at 2000 × g for 5 min at 4 °C. Supernatant was removed and the cells were resuspended in 500 μL of 0.9X PBS and kept on ice.

**ATAC-Seq**. 50,000 cells per sample underwent tagmentation reaction with Nextera Tn5 transposase using the Illumina Nextera kit and purified with a Qiagen MinElute reaction kit[48]. The DNA was then PCR amplified for 9 cycles to add indexing primers. SPRI AMPure beads enriched for fragments under ~600 bps. The DNA was cycled again with the 9-cycle protocol, followed by cleanup with SPRI AMPure beads. The DNA was quantified with a Qubit DNA High Sensitivity assay and analyzed for quality and size distribution on an Agilent TapeStation with a

High Sensitivity D5000 ScreenTape. Samples were pooled for a 10 nM final overall concentration. Sequencing was performed with an Illumina HiSeq 2500 system with 2 × 50 bp read length by the Washington University in St. Louis School of Medicine Genome Technology Access Center. ATAC-Seq data are available on GEO under GSE145551.

**ATAC-Seq data analysis**. Reads were demultiplexed then trimmed with Cutadapt (adaptor sequence 5′-CTGTCTCTTATACACATCT-3′ for both reads) and checked with FastQC to ensure quality. The reads were aligned to the GRCz10/danRer10 genome using BWA-MEM and sorted using SAMtools. Duplicate reads removed with Picard tools using the following parameters: ASSUME_-SORTED=true, VALIDATION_STRINGENCY=LENIENT. The files were indexed with SAMtools then filtered for high-quality alignments using the following parameters: -f 3, -F 4, -F 8, -F 256, -F 1024, -F 2048, -q 30. MACS2 was then used to identify peaks with the callpeak command with parameters -g 1.4e9, -q 0.05, --nomodel, --shift −100, --extsize 200. Differential peak accessibility was assessed with the MACS2 bdgdiff function. HOMER was then used to annotate the peaks (annotatePeaks.pl).

**Cloning putative enhancers**. All putative enhancer elements except for *peak13* were PCR amplified from genomic DNA from AB* zebrafish larvae or *Tg* (*BRAF^{V600E}*);*p53^{−/−}* larvae with high-fidelity DNA polymerases. *peak13* was amplified from BAC CHORI-211:212I14 (a gift from the Johnson Lab). Genomic locations for each putative enhancer in the Zv10 build of the zebrafish genome are listed in Supplementary Table 2 and primers used to amplify each sequence and total cloned sizes of each element are listed in Supplementary Table 3. PCR amplified enhancer elements were cloned into pENTR5′ vectors (ThermoFisher). Anytime Phusion (NEB) was utilized to amplify a putative enhancer, adenines were added to the end of the PCR product using *Taq* polymerase (Promega) before performing a TOPO reaction. Using plasmids from the Tol2 kit, all putative enhancers were placed upstream of a basal promoter, mouse *beta-globin*[27] via Gateway LR reactions with the following combination of Gateway vectors: p5E-enhancer, pME-*betaglobin:EGFP*, p3E-polyA, and pDestTol2pA. To generate a *sox10* minimal promoter plasmid, a *sox10* minimal promoter was PCR amplified from genomic DNA from AB* larvae with primers (Supplementary Tables 2 and 3) containing overlapping nucleotides to sequences on either side of the AgeI site in pENTR-EGFP2 (Addgene #22450), similarly described in a previous study[49]. The PCR product was then cloned into pENTR-EGFP2 and digested with AgeI, through a Gibson Assembly reaction using an in-house Gibson mixture (gifted by the Solnica-Krezel lab, Washington University in St. Louis; NEB Phusion, NEB Taq DNA Ligase, and NEB T5 Endonuclease). This pME-*sox10min:EGFP* was then used in a Gateway LR reaction with p5E-MCS, p3E-polyA, and pTol2Destp2A to generate the final reporter construct. All PCR amplified sequences were Sanger sequenced by Genewiz. The list of plasmids used and generated in this study can be found in Supplementary Table 4. The conserved *peak5* sequence was PCR amplified from the previously cloned *peak5* sequence, cloned into a pENTR5′ vector, and placed upstream of *betaglobin:EGFP* through a Gateway reaction, as described above.

**Screening putative enhancers and generation of transgenic zebrafish lines**. One cell stage *Tg(BRAF^{V600E})*;*p53^{−/−}* embryos were injected with 20 ng/μL of an enhancer reporter plasmid in addition to 15 ng/μL of *crestin:mch*, and 20 ng/μL of Tol2 transposase mRNA. 1 dpf larvae were screened for EGFP and mCh localization in the neural crest on a stereomicroscope. Embryos were scored as "strong expression" if 5 or more cells were EGFP and mCh positive. Three technical and biological replicates of injections were performed for each putative enhancer. Two technical replicates and three technical replicates were performed for Negative Control A and Negative Control B, respectively. At least 30 embryos scored as positive for each putative enhancer were raised to adulthood. Founders were identified by out-crossing F0 adults, and progeny were screened from 1 to 5 dpf for EGFP localization and subsequently raised to adulthood. Lines screened for *EGFP* expression in melanoma patches and tumors are listed in Supplementary Table 1.

**Imaging**. For most live-imaging, larvae were anesthetized in Tricaine, embedded in 0.8% agarose, and imaged with a Nikon SMZ18 fluorescent dissecting microscope. Higher resolution of *peak5* transgenic embryos at 1 dpf was performed with a Zeiss Lightsheet 7 in the Washington University Center for Cellular Imaging (WUCCI). Embryos were dechorionated by hand and placed in 1.15% low-melt agarose, diluted from 1.5% with 4% Tricaine, in a glass capillary tube. Each embryo embedded in agarose was slightly extruded from the capillary tube for imaging. For imaging adults, fish were anesthetized in Tricaine and imaged with a Nikon SMZ18 fluorescent dissecting microscope. Images were processed with Photoshop and ImageJ. The Photomerge function in Photoshop was used to stitch together tiled images of adult zebrafish.

**Evolutionary conserved sequence identification**. To generate the Dot-Matrix view, sequences surrounding the *sox10* locus in both zebrafish (Chromosome 3: 1976387-2026723, danRer10) and carp (scaffold: LG6, Chromosome 6: 16126287-16150639), were aligned with optimization for more dissimilar sequences in BLAST (http://blast.ncbi.nlm.nih.gov). BLAST with optimization for more

dissimilar sequences with discontiguous megablast was also used to compare the zebrafish *peak5* sequence to selected members of the *Cyprinidae* family (Supplementary Table 5). The nucleotide collection or whole-genome shotgun contigs databases were chosen as the search set. The most conserved regions of *peak5* nucleotide sequences from each species were then aligned with M-Coffee (http://tcoffee.crg.cat/apps/tcoffee/do:mcoffee). The fasta_aln file was downloaded and Boxshade (https://embnet.vital-it.ch/software/BOX_form.html) was then used to shade the alignment using RTF_new as the output format and "other" as the input sequence format. TFBS predictions were identified with FIMO (http://meme-suite.org/tools/fimo) using the JASPAR core 2016 vertebrate list. SoxE dimeric binding sites were predicted with JASPAR (http://jaspar.genereg.net/search?q=&collection=CORE&tax_group=vertebrates), searching explicitly for Sox10 motifs using an 80% relative profile score threshold.

**Plasmid mutagenesis and deletion analysis**. The Q5 Site-Directed Mutagenesis Kit (NEB) was used according to the manufacturer's instructions to either delete the *peak5* conserved 192 bp sequence or mutate SoxE binding sites. The correct deletion or mutations were confirmed through Sanger Sequencing at Genewiz. 20 ng/μL of *peak5* full-length wild-type, *peak5Δ192*, *peak5_conserved* or *peak5* full-length SoxE site mutated plasmids were co-injected with 15 ng/μL of *crestin:mch* and 20 ng/μL of Tol2 transposase mRNA into embryos derived from a *Tg* (*BRAF^{V600E}*);*p53^{−/−}* in-cross. Only embryos that expressed *mCh*, ensuring successful injection, were scored on 1 dpf for expression of *EGFP*. EGFP positive embryos that had identifiable cell-type labeling were categorized into the following cell-localization categories: neural crest, KA neurons, both the neural crest and KA neurons, or ectopic. The scorer was blinded to treatment groups when scoring. Three technical and biological replicates were performed for the conserved deletion experiment and two technical and biological replicates were performed for the SoxE TFBS mutation experiment. Stable lines for *peak5Δ192* (Supplementary Table 1), *peak5_conserved*, and *peak5_SoxEmut* were identified and characterized as described above.

**sgRNA transcription**. sgRNAs were designed using CHOPCHOP (http://chopchop.cbu.uib.no) or the ZebrafishGenomics hub[50] on the UCSC Genome Browser. 20 bp targets were chosen based upon ranking and if they began with either GG or GA nucleotides. A single nucleotide was mutated if the chosen sgRNA did not begin with GG or GA (see Supplementary Table 6). sgRNAs were then assembled by PCR similar to previously described methods[50,51]. Briefly, sgRNA oligonucleotide sequences were ordered from IDT with the following composition: GG beginning sgRNA oligonucleotides were placed downstream of a T7 promoter sequence (5′-TAATACGACTCACTATA-3′) and GA beginning sgRNAs oligonucleotides were placed downstream of a SP6 promoter sequence (5′-ATTTAGGT-GACACTATA-3′). After each sgRNA sequence, an oligonucleotide sequence complimentary to a reverse-compliment tracrRNA sequence oligonucleotide was added (5′-GTTTTAGAGCTAGAAATAGC-3′). PCR using Phusion (NEB) was performed with the sgRNA sequence containing oligonucleotide and the reverse-compliment tracrRNA sequence oligonucleotide (5′-AAAAGCACCGACTCGGTG CCACTTTTTCAAGTTGATAACGGACTAGCCTTATTTTAACTTGCTATTTC TAGCTCTAAAAC-3′), ordered from IDT, to generate dsDNA for subsequent transcription. PCR product was purified with a QIAquick PCR Purification Kit (Qiagen). PCR purified product was then transcribed using either a T7 MEGA-script (Ambion) kit or a SP6 MAXIscript kit (Invitrogen). sgRNAs were DNaseI treated and purified with Micro Bio-Spin^{TM} P-30 Gel Columns (BioRad). RNA quality was verified by gel electrophoresis and concentrations were measured by a Nanodrop.

**CRISPR injections and genotyping**. Three pooled sgRNAs for *sox10min* were injected at concentrations ranging from 59 to 94 ng/μL with ~3.125 μM/μL of EnGen Cas9 NLS, *S. pyogenes* (NEB) and phenol red into *Tg(BRAF^{V600E}*;*crestin: EGFP)*;*p53^{−/−}* single-cell embryos. Four pooled *peak5* sgRNAs were injected at concentrations ranging from 11.25 to 75 ng/μL with ~3.22 μM/μL of EnGen Cas9 NLS, *S. pyogenes* (NEB) and 1.6X Cas9 Nuclease Buffer (NEB) into *Tg(BRAF^{V600E})*;*p53^{−/−}* single-cell embryos. At 1 dpf, embryos were removed from their chorions and DNA was extracted via the HotSHOT method[52]. PCR of the target region was performed using genomic DNA from 8 control (uninjected or Cas9 only injected) embryos and 8 CRISPR injected embryos using primers listed in Supplementary Table 7. A CRISPR was deemed effective if DNA laddering was observed in the CRISPR injected embryos compared to a single WT band in the control embryos (Fig. 6a, b). F0 embryos were grown to maturity and outcrossed to identify deletions transmitted through the germline. Mutant bands were gel extracted using QIAquick Gel Purification Kit (Qiagen), cloned into a PCR2.1-TOPO vector or PCR4-TOPO vector (Invitrogen), and Sanger sequenced at Genewiz. Primers to genotype each *sox10min* and *peak5* deletion alleles are listed in Supplementary Table 7. To definitively distinguish heterozygotes from mutants, a second geno-typing assay for both *sox10min* and *peak5* alleles was utilized and is listed in Supplementary Table 7. Example gels for each genotyping assay are displayed in Supplementary Fig. 9b-e.

**Whole-mount in situ hybridization**. 1 dpf embryos from a *sox10min*[stl792/+] in-cross and *peak5*[stl538/+] in-crosses were dechorionated and fixed in 4% paraformaldehyde. Embryos were then dehydrated with 100% MeOH (5 × 5 min washes) and stored at −20 °C. In situ hybridization for *sox10* mRNA was then performed as previously described[53]. Embryos were then scored blinded to genotype based on *sox10* expression levels. Embryos from the *sox10min*[stl792/+] in-cross were scored as either "strong," "reduced," or "absent" *sox10* expression. Embryos from *peak5*[stl538/+] in-crosses were scored as either "strong," "slightly reduced," or "reduced" *sox10* expression. Scored embryos were imaged and then processed for genotyping, as described above. One biological replicate and two technical replicates were performed for *sox10min* in situ analysis. Two biological replicates and five technical replicates were performed for *peak5* in situ analysis.

**Adult phenotyping**. To analyze and quantify adult the *peak5*[stl538] phenotype, both sides of 20 *Tg(BRAF*[V600E]*);p53*[−/−]*;peak5*[+/+] and 20 *Tg(BRAF*[V600E]*);p53*[−/−]*; peak5*[stl538/stl538] fish were imaged and the number of complete stripe breaks on each fish was counted. Lateral images from both sides of 20 *Tg(BRAF*[V600E]*);p53*[−/−]*; peak5*[+/+] and 20 *Tg(BRAF*[V600E]*);p53*[−/−]*;peak5*[stl538/stl538] 3 mpf adult fish were captured with a Nikon SMZ18 dissecting microscope. With the scorer blinded to genotype, the number of complete stripe breaks was manually counted. To analyze the adult *Tg(BRAF*[V600E]*);p53*[−/−]*;peak5*[stl538/+]*;sox10*[stl792/+] phenotype, 10 2 mpf fish from each genotype, derived from the same in-cross, were analyzed in the same aforementioned manner.

**Statistics and reproducibility**. A Chi-squared analysis performed to analyze the *peak5* conserved sequence deletion mutagenesis experiment in GraphPad Prism 8 (Fig. 4d). Fisher's exact tests (https://astatsa.com/FisherTest/) were used to analyze the SoxE TFBS mutagenesis injection experiment (Fig. 5d) as well as in situ phenotypes for *sox10* mRNA expression in wild-type, heterozygous, and mutant *sox10min*[stl792] and *peak5*[stl538] embryos (Fig. 6g, j). Welch's *t*-tests were performed in GraphPad Prism 8 to analyze the number of stripe breaks in adult fish across genotypes (Fig. 6m, r). Sample numbers (*n* numbers) are in the main text or figure/figure legends, and replicates are described in figure legends and in relevant parts of methods above in detail.

**Reporting summary**. Further information on research design is available in the Nature Research Reporting Summary linked to this article.

## Data availability
ATAC-Seq data are available on GEO under GSE145551, and other source data is available from the corresponding author on reasonable request. Numerical source data for quantification of EGFP reporter expression (Figs. 4d, 5d), *sox10* expression (Fig. 6g, j), and stripe breaks per animal (Fig. 6m, r) are available in Supplementary Data 1.

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

## Acknowledgements

Research reported in this publication was supported in part by the National Cancer Institute of the National Institutes of Health under award number R01CA240633. The content is solely the responsibility of the authors and does not necessarily represent the official views of the National Institutes of Health. R.L.C. was supported by the National Science Foundation Graduate Research Fellowship (DGE-1745038). C.K.K. was supported by the Cancer Research Foundation Young Investigator Award. E.T.K. was supported by T32 GM007067. S.K.D. was supported by T32 GM007067. P.M.G. was supported by the National Science Foundation Graduate Research Fellowship (DEG-1745038). We thank members of the Kaufman lab, Souroullas lab (Washington University in St. Louis), and Petersen lab (Kenyon College) for helpful discussions; the Washington University Zebrafish Consortium; the Washington University in St. Louis School of Medicine Genome Technology Access Center; Bo Zhang in the Department of Developmental Biology and the Center for Regenerative Medicine at Washington University School of Medicine for assistance with ATAC-Seq analysis; P. Bayguinov in the Washington University Center for Cellular Imaging; S. Johnson lab members for the BAC plasmid (Washington University in St. Louis); S. Kucenas (University of Virginia) for the Tg(sox10:mRFP) line; M. Bagnall (Washington University in St. Louis) for identifying the Kolmer–Agduhr neurons; M. Mokalled (Washington University in St. Louis) for the CRISPR protocol; M. Duncan (University of Kentucky) for statistical analysis guidance.

## Author contributions

R.L.C., E.T.K., and C.K.K. designed research, and R.L.C., E.T.K., S.K.D., P.M.G., A.P.Z., V.G., and S.S. performed research. R.L.C. and C.K.K. analyzed data, and R.L.C. and C.K.K. wrote the manuscript.

## Competing interests

The authors declare no competing interests.
