## [Transparent Peer Review File · Communications Biology]

Reviewers' comments:

Reviewer #1 (Remarks to the Author):

The study by Cunningham and colleagues interrogates the regulatory landscape of the neural crest transcription factor Sox10 in a zebrafish melanoma model. They aim at identifying enhancer elements shared between neural crest and melanoma cells. By combining ATAC-seq and an in vivo reporter system, the authors identified a region around the Sox10 locus termed as "peak5", which can drive transgene GFP expression in zebrafish neural crest and is also active in melanoma tumors and early oncogenic lesions.

The authors dissected peak5 region to identify a conserved 192 bp region necessary for enhancer activity in both the cranial neural crest and melanoma cells. Further bioinformatic analysis of this conserved element revealed two dimeric SoxE sites, which were essential for enhancer activity. These results led the authors to propose that the peak5 enhancer element promotes Sox10 expression in neural crest and early melanoma by a feed-forward mechanism wherein Sox10 promotes its own expression.

The identification of an enhancer element of Sox10 shared between neural crest and melanoma lesions in vivo is indeed interesting and further highlights mechanisms by which the neural crest transcriptional program is re-activated during melanoma progression. Yet, the manuscript has a number of limitations that should be addressed before being considered for publication. These include issues with the imaging of transgenic fish, a need for additional quantitative analysis, and lack of evidence showing the necessity of the peak5 element for Sox10 expression.

Major concerns:

- Central to the authors' model is the assumption that peak5 is one of the primary enhancers of Sox10 in the neural crest. Yet there is no direct evidence that peak5 is necessary Sox10 expression. The only tangential data provided are images of double transgenic zebrafish embryos injected with peak5:betaglobin:Egfp and Sox10:mRFP - where the authors attempt to show that the enhancer element is active in the cells expressing Sox10 (Fig3b). These images are not sufficient to support this claim since most of the Sox10:mRFP positive cells do not have transgene GFP expression. The authors should perform functional experiments to establish that peak5 activity is critical of Sox10 expression. One such strategy could be targeting the enhancer element with CRISPR/CAS9 and asses Sox10 expression in neural crest and melanoma tumors in zebrafish.
- The images of the peak5:beta-globin: EGFP throughout the figures is variable, with the transgene appearing to be distinct display patterns between embryos. This is a significant issue in the experiments that show peak5 activity in the cranial neural crest. Even in the stable lines, peak5 activity is only observed only in LineA and Line129B at 1dpf. Though the authors claim that the variation in integration position could account for this, it is still reason for concern - as one of the phenotypes scored following enhancer manipulation is the loss of activity in the cranial neural crest.
- Related to the point above, the overlap between peak5 expressing cells and crestin-positive cells is also highly variable. From the representative images, only a small percentage of cells appear to express both reporters. To quantitatively assess the overlap, the authors may want to perform FACS analysis to measure the number of crestin/peak5 dual positive cells in the embryo (the same analysis might also be informative if performed in melanoma lesions).
- The authors show that the SoxE site mutant peak5 element is not expressed in pre-migratory neural crest (Figure 6), but they do not test if this mutation disrupts enhancer activity in melanoma tumors. This experiment should be included in the study since the proposed model states that the Sox10 feedback loop is shared between neural crest and melanoma.

- Imaging of transgenic embryos throughout the paper should be improved. The low magnification images are not sufficient to confirm specific enhancer activity in specific cell populations, and the pictures of whole-body adult fishes are difficult to interpret. Reporter fluorescence is often indistinguishable from autofluorescence in the adult fish.

Reviewer #2 (Remarks to the Author):

Reviewer comments:

Cunningham et al. investigate the regulatory inputs required of sox10 expression in melanoma using the melanoma prone Tg(BRAFV600E;crestin:EGFP);p53^{-/-} zebrafish line.

Key observations from this manuscript include:

- (1) the identification of regulatory elements near sox10 that are active in a subset of neural crest cells,
- (2) demonstration that one such enhancer (peak 5) is active in zebrafish melanoma-initiating and melanoma cells,
- (3) and that peak 5 is sensitive to the loss of SOXE binding sites centred within this element.

The current manuscript is well-written and the experiments are sound. It is noteworthy that in a previous paper, Kaufman et al. reported that melanoma derives from cells with neural crest properties thus it is expected that enhancers active in neural crest will be active in melanoma.

Minor concerns

While not a condition of acceptance (in my view), I think the current manuscript would be enhanced by reporting the expression patterns (active or inactive) of the additional elements, shown to have neural crest activity (other than peak 5), in melanoma. It would be nice to compare the enhancer properties of all peaks with peak 5 in order to show the specificity of peak 5 among the other enhancers in melanoma.

The title should include "putative sox10 enhancers" as this study reports tissue specificity of the tested enhancers and does not directly study enhancer – promoter contacts.

The ATAC-seq datasets generated on melanoma cell lines and tumor samples are a valuable addition to the field and should be deposited to a public database. There is no mention of whether these datasets will be made publicly available.

Figure 1b: What is the rationale for inclusion of Peak 7 in this study? This region looks "closed" inactive. Figure 1b would benefit from including the MACS2 called peak bed file as a separate track to show the reader that these peaks are significantly enriched above background.

Scale bars representing signal strength for each track should be added.

Figure 1S would benefit from including a track representing neural crest ATAC-seq data for comparison with tumor and cell line samples (taken if possible, with publisher's permission, for instance from Chong-Morrison et al., 2018; Lukoseviciute et al., 2018).

Moreover, what exactly do the red bars represent? If these are regions of called peaks by MACS2, it should be mentioned in the legend. The current description "Red bars indicate potential peaks" is not

an adequate description.

Figure 1d-f: It is difficult to see the embryo with the black background. In Figure 6e-j the authors do a good job of outlining the embryo, please repeat this in Figures 1d-f.

Figure 1d: It is very difficult to see the merge/ overlap of peak-5 expressing cells and sox10-rfp positive expressing cells in the current image.

Page 7, line 12: The authors mention that 9 of 11 putative enhancer had NCC expression. "Nine out of eleven putative enhancers drove EGFP expression in at least a subset of NCCs" Peak 5 showed the most robust signal and the authors continued to explore this enhancer in a tumor setting, however, it would interesting to report whether the other enhancers that are active in NC and are also active melanoma as this could help determine the specific sox10 regulatory element in transformed cells.

Page 45: Table 1 reports annotated peak location, amplified location and distance from sox10 TSS. When using the primers listed in Table 2 for each peak, I find discrepancies between the amplified peak location mentioned for Peak 2 and Peak 7 in Table 1 and the actual amplified peak location when using the function "in-silico PCR" on the UCSC genome browser (DanRer10). For example, Table 1 reports the location of Peak 2 to be Chr3: 1977785-1978218, however, using the primes given for Peak 2, the genome location is in fact Chr3: 1977258-1978528.

Page 46: Table 2 "Notes" column reports the number of bps amplified for each peak. It is not clear what the meaning of this column is as when using the primers given in this table the amplified product is large than stated for all all peak. For example, the amplified product for peak 2 is as 434bps, however, the using the primers given the product size and based on the genome coordinates the product size should be 1271bps.

Page 9, line 1-4: " peak 5 transgenic lines express EGFP within a subset of ventral Kolmer-Agdhur (KA) neurons in the spinal cord, which contact cerebrospinal-fluid, as identified by cell shape and localization. By 5 dpf, EGFP expression was consistently observed in KA neurons across all lines, and in the heart,....." Is sox10 expressed in KA neurons and cells within heart or is this thought to be ectopic activity of the peak 5 enhancer?

Page 10, line 13: Control peaks1 and peak 8 are said to be embryonically expressed, does this exclude neural crest activity?

Reviewers' comments:

Reviewer #1 (Remarks to the Author):

The study by Cunningham and colleagues interrogates the regulatory landscape of the neural crest transcription factor Sox10 in a zebrafish melanoma model. They aim at identifying enhancer elements shared between neural crest and melanoma cells. By combining ATAC-seq and an in vivo reporter system, the authors identified a region around the Sox10 locus termed as “peak5”, which can drive transgene GFP expression in zebrafish neural crest and is also active in melanoma tumors and early oncogenic lesions.

The authors dissected peak5 region to identify a conserved 192 bp region necessary for enhancer activity in both the cranial neural crest and melanoma cells. Further bioinformatic analysis of this conserved element revealed two dimeric SoxE sites, which were essential for enhancer activity. These results led the authors to propose that the peak5 enhancer element promotes Sox10 expression in neural crest and early melanoma by a feed-forward mechanism wherein Sox10 promotes its own expression.

The identification of an enhancer element of Sox10 shared between neural crest and melanoma lesions in vivo is indeed interesting and further highlights mechanisms by which the neural crest transcriptional program is re-activated during melanoma progression. Yet, the manuscript has a number of limitations that should be addressed before being considered for publication. These include issues with the imaging of transgenic fish, a need for additional quantitative analysis, and lack of evidence showing the necessity of the peak5 element for Sox10 expression.

Major concerns:

- Central to the authors' model is the assumption that peak5 is one of the primary enhancers of Sox10 in the neural crest. Yet there is no direct evidence that peak5 is necessary Sox10 expression. The only tangential data provided are images of double transgenic zebrafish embryos injected with peak5:betaglobin:EGFP and Sox10:mRFP - where the authors attempt to show that the enhancer element is active in the cells expressing Sox10 (Fig3b). These images are not sufficient to support this claim since most of the Sox10:mRFP positive cells do not have transgene GFP expression.

We agree with the limitations of the original images in evaluating the overlap of peak5-driven EGFP and sox10 transgene-driven mRFP. To address this, we re-imaged peak5 transgenic embryos (in which EGFP is not restricted in distribution within the nucleus and cytoplasm) and sox10:mRFP (in which RFP is membrane bound) using Zeiss light sheet microscopy, and show a high degree of co-labeling in both representative dorsal and lateral views (Figure 2d-e) in posteriorly-located premigratory neural crest (the neural crest cell population in peak5 transgenes we analyzed throughout the manuscript). In addition, we have added insets in Figure 1d-f showing enlargements and labeling to of the dorsal/caudal area to better highlight reporter expression.

The authors should perform functional experiments to establish that peak5 activity is critical of Sox10 expression. One such strategy could be targeting the enhancer element with CRISPR/CAS9 and assess Sox10 expression in neural crest and melanoma tumors in zebrafish.

We agree that modifying the endogenous enhancer locus in the genome (e.g. by deleting the region) is the most direct, if not technically challenging, approach to demonstrate the function of a given region such as *peak5*. To address this as suggested, we used CRISPR/Cas9 to delete either the *sox10* promoter region (including the transcriptional start site, referred to as *sox10* minimal promoter region) and *peak5* including the most evolutionarily conserved sequence.

The *sox10* minimal promoter region deletion resulted in complete abrogation of *sox10* expression as determined by whole mount *in situ*, a phenotype even more severe than previously reported for *sox10* mutant alleles (i.e. *colourless* mutant) (Fig. 6). Further, the *peak5* deletion led to reduction, but not complete abrogation of, *sox10* expression by whole mount *in situ* as well in embryos (Figure 6). In the adult, even heterozygous loss of *sox10* expression in the *sox10* minimal promoter led to disruption of normal pigmentation/melanocyte stripe formation, a phenotype consistent with *sox10* abnormalities in mammals and humans in some types of Waardenburg syndrome (Figure 6). Similarly, homozygous deletion of *peak5* also led to a stripe defect (Figure 6). Together, we believe this strongly supports that *peak5* is a *bona fide* transcriptional enhancer for *sox10* that is necessary for normal *sox10* expression during development and for proper specification or patterning of pigmented cells/melanocytes which are known to require proper *sox10* function.

While we agree that determining the potential role of *peak5* in melanoma onset will be illuminating, we believe this aspect of study will require additional long term study in future efforts given the lengthy time required for tumor formation (median onset of ~ 20 weeks) in addition to determining if this genetic perturbation results in an even further delay in tumor onset. In addition, the apparent disruption of normal melanocyte formation/stripe patterning raises the possibility that fewer melanocytes are present in *peak5* deletion homozygotes to produce tumors from the outset, which will further complicate analysis of tumor onset rates and require new approaches to this analysis (e.g. better normalizing for melanocyte number per fish).

- The images of the *peak5*:beta-globin:EGFP throughout the figures is variable, with the transgene appearing to be distinct display patterns between embryos. This is a significant issue in the experiments that show *peak5* activity in the cranial neural crest. Even in the stable lines, *peak5* activity is only observed only in LineA and Line129B at 1dpf. Though the authors claim that the variation in integration position could account for this, it is still reason for concern - as one of the phenotypes scored following enhancer manipulation is the loss of activity in the in the cranial neural crest.

We agree that the variability amongst *peak5* transgenes could suggest that it is also active in cranial neural crest. As *sox10* is active in non-neural crest cell populations, such as the CNS, we chose to focus on the posterior dorsally located premigratory NCCs for assessing *peak5* activity in all of our experiments as this cell population is most likely to be *sox10* expressing NCCs (and was confirmed through Lightsheet microscopy, Fig 2e,f). The presence of the posterior dorsally located premigratory NCCs is present in Lines A, B, (Figure 2 a,b) 129B, and 219K (Figure S4 a,b).

- Related to the point above, the overlap between *peak5* expressing cells and crestin-positive cells is also highly variable. From the representative images, only a small percentage of cells appear to express both reporters. To quantitatively assess the overlap, the authors may want to perform FACS analysis to measure the number of crestin/*peak5* dual positive cells in the embryo (the same analysis might also be informative if performed in melanoma lesions).

We agree it remains an interesting question of the relationship between crestin and sox10 labeling of neural crest, and in the case of sox10, potentially other lineages in the central nervous system. As noted above, we focused efforts to better analyze the expression pattern of the peak5 region of the sox10 upstream region in relation to an existing published sox10 reporter given the limitations of our original images in evaluating the overlap of peak5-driven EGFP and sox10 transgene-driven mRFP in developing embryos from stable transgenic lines. To address this, we re-imaged peak5 transgenic embryos (in which EGFP is not restricted in distribution within the nucleus and cytoplasm) and sox10:mRFP (in which RFP is membrane bound) using Zeiss light sheet microscopy, and show a high degree of colabeling in both representative dorsal and lateral views (Figure 2d-e). We agree that a more complete accounting of the relationship between crestin reporter expression and existing sox10 transgenes and our own peak5 transgenics, most importantly in relation to melanoma onset, could be interesting in the future addressing questions related to: e.g. does peak5 reporter expression precede crestin reporter expression?

- The authors show that the SoxE site mutant peak5 element is not expressed in pre-migratory neural crest (Figure 6), but they do not test if this mutation disrupts enhancer activity in melanoma tumors. This experiment should be included in the study since the proposed model states that the Sox10 feedback loop is shared between neural crest and melanoma.

We agree this is a fascinating question, and we are happy to report that, as noted, not only does mutation of the SoxE binding sites abrogate transgene reporter expression in embryos, but also substantially reduces EGFP reporter expression in the bulk of melanoma tumors that form (Figure 5m, 11/13 tumors from 11 different animals from two different stable transgenic lines with mutated SoxE binding sites show no detectable EGFP expression). We find this result remarkable that such a small change in the context of a 669 bp enhancer can lead to detectable decrease in reporter activity in the tumor context, and overall, believe this better supports the model in Figure 6s.

- Imaging of transgenic embryos throughout the paper should be improved. The low magnification images are not sufficient to confirm specific enhancer activity in specific cell populations, and the pictures of whole-body adult fishes are difficult to interpret. Reporter fluorescence is often indistinguishable from autofluorescence in the adult fish.

We have attempted to address this concern by adding insets of key portions of the embryos (e.g. Fig 1d-f) and additional of light sheet imaging (Fig 2d-e), as noted above as well. For adults, we would highlight the images in Figure 3 that tumors arise on the dorsal body of the adult fish (e.g. Fig 3a', d', e', g, h) where minimal autofluorescence is present or on the tail where the autofluorescent interstripe (asterisks in Fig 2f) is more readily differentiated from the tumor fluorescence (Fig 3b', c', f, i, j).

Reviewer #2 (Remarks to the Author):

Reviewer comments:

Cunningham et al. investigate the regulatory inputs required of sox10 expression in melanoma using the melanoma prone Tg(BRAFV600E;crestin:EGFP);p53^{-/-} zebrafish line.

Key observations from this manuscript include:

- (1) the identification of regulatory elements near sox10 that are active in a subset of neural crest cells,
- (2) demonstration that one such enhancer (peak 5) is active in zebrafish melanoma-initiating and melanoma cells,
- (3) and that peak 5 is sensitive to the loss of SOXE binding sites centred within this element.

The current manuscript is well-written and the experiments are sound. It is noteworthy that in a previous paper, Kaufman et al. reported that melanoma derives from cells with neural crest properties thus is it expected that enhancers active in neural crest will be active in melanoma.

Minor concerns

While not a condition of acceptance (in my view), I think the current manuscript would be enhanced by reporting the expression patterns (active or inactive) of the additional elements, shown to have neural crest activity (other than peak 5), in melanoma. It would be nice to compare the enhancer properties of all peaks with peak 5 in order to show the specificity of peak 5 among the other enhancers in melanoma.

We agree that, in principle, additional study of the reporter expression patterns for each of the other enhancer elements will be illuminating. While the relatively medium throughput analysis of multiple reporter transgenes in F0 embryos is a strength of this zebrafish system and approach, the variability of individual embryo expression patterns, as noted in other Reviewers comments, (which can be somewhat addressed with high numbers of replicates) ultimately seems to require the generation of multiple stable transgenic lines for each element under study, as we did for peak5, sox10min promoter, and peak 8. This proves to be a laborious and low-throughput undertaking, and in the course of revisions, we found that the CRISPR deletion of peak5 and the sox10 minimal promoter region was, perhaps, most instructive in establishing the endogenous function of the peak5 region in regulating sox10 expression and neural crest/pigment cell formation. We agree a comparison of the various enhancer properties will be fascinating and expect the way forward in future studies will be to generate multiple deletion strains, one for each enhancer, then even allowing for combinatorial analysis of trans-heterozygote phenotypes with combinations of different enhancers, as we did with the peak5 and sox10 minimal promoter deletions, which together produce an additive effect in increasing the stripe patterning defect (Fig 6).

The title should include “putative sox10 enhancers” as this study reports tissue specificity of the tested enhancers and does not directly study enhancer – promoter contacts.

We appreciate the recommended title change to better reflect the original work and have adjusted accordingly. With the addition of the CRISPR-deletion of peak5 showing an embryonic (diminished sox10 expression) and adult (stripe patterning defect) in Figure 6, we wonder if this

additional data would better satisfy the original title in regards to peak5, and are happy to adjust wording to best reflect the consensus.

The ATAC-seq datasets generated on melanoma cell lines and tumor samples are a valuable addition to the field and should be deposited to a public database. There is no mention of whether these datasets will be made publicly available.

We very much appreciate this comment noting the potential utility of these datasets to the community and absolutely want to share with the wider community. We apologize for the oversight in not including the GEO accession information and have added in the text that data will be available at publication under GSE145551 (page 23).

Figure 1b: What is the rationale for inclusion of Peak 7 in this study? This region looks “closed” inactive. Figure 1b would benefit from including the MACS2 called peak bed file as a separate track to show the reader that these peaks are significantly enriched above background.

We agree Peak 7 has somewhat variable accessibility across the multiple tumor samples (Supplemental 1A). We include the called peak regions in Supp 1A (red bars indicating extent of called significant peaks by MACS2), and note that peak 7 is called present in 4 of the melanoma tumors in this study and both of the melanoma cell lines from Kaufman et al, 2016. We have better labeled Supp Fig 1A as well to identify the peaks under study.

Scale bars representing signal strength for each track should be added.

We have added signal strength bars in our reanalysis of the data in comparison to other ATAC-Seq data as suggested in Supp 1B, allowing better comparison between peaks within and between each experiment.

Figure 1S would benefit from including a track representing neural crest ATAC-seq data for comparison with tumor and cell line samples (taken if possible, with publisher’s permission, for instance from Chong-Morrison et al., 2018; Lukoseviciute et al., 2018).

We very much appreciate this suggestion and have added analysis of published neural crest ATAC-Seq data as indicated in a new Figure Supplemental 1B. We found it fascinating and reassuring that the general location of accessible domains near *sox10* becomes apparent by 5-6 somite stage and persists into later stages in *sox10*-expressing cells.

Moreover, what exactly do the red bars represent? If these are regions of called peaks by MACS2, it should be mentioned in the legend. The current description “Red bars indicate potential peaks” is not an adequate description.

We have clarified this in the figure legend for Supplemental Figure 1A and apologize for this oversight.

Figure 1d-f: It is difficult to see the embryo with the black background. In Figure 6e-j the authors do a good job of outlining the embryo, please repeat this in Figures 1d-f.

We added the embryo outlines as suggested.

Figure 1d: It is very difficult to see the merge/ overlap of peak-5 expressing cells and *sox10*-rfp positive expressing cells in the current image.

We have attempted to address this in two ways – by adding insets in Figure 1d-f to better highlight the expression, and by including the Zeiss light sheet imaging in Figure 2d-e showing peak5:EGFP and sox19:mRFP colabeling at 24 hpf.

Page 7, line 12: The authors mention that 9 of 11 putative enhancer had NCC expression. “Nine out of eleven putative enhancers drove EGFP expression in at least a subset of NCCs” Peak 5 showed the most robust signal and the authors continued to explore this enhancer in a tumor setting, however, it would interesting to report whether the other enhancers that are active in NC and are also active melanoma as this could help determine the specific sox10 regulatory element in transformed cells.

We agree the differential behavior of individual enhancer elements in driving transgene expression is interesting and, we believe, will lead to a better understanding of the complexity of sox10 gene regulation. As suggested, we note that while *peak1* and *peak8* broadly showed expression in fin mesenchyme and the CNS, respectively, in stable transgenic embryos (page 10, Fig S2), these reporters did not show expression in adult melanoma tumors (Supp S6), supporting that *peak5* has specificity for the melanoma state. To get a full accounting of all enhancers’ activity in melanoma tumors will require generation of multiple independent stable transgenic reporters for each region and development of tumors in these lines, which becomes a large undertaking beyond the scope of this current analysis, we believe.

Page 45: Table 1 reports annotated peak location, amplified location and distance from sox10 TSS. When using the primers listed in Table 2 for each peak, I find discrepancies between the amplified peak location mentioned for Peak 2 and Peak 7 in Table 1 and the actual amplified peak location when using the function “in-silico PCR” on the UCSC genome browser (DanRer10). For example, Table 1 reports the location of Peak 2 to be Chr3: 1977785-1978218, however, using the primes given for Peak 2, the genome location is in fact Chr3: 1977258-1978528.

We are so grateful for the note of this mistake and the diligence in reviewing this table. We have updated the primers for peak 2 and peak 3 which had been inadvertently combined. We confirmed the location of peak7 and the amplified sequence used in our analysis matches what we reported initially, but we again very much appreciate this query as it appears that the in-silico PCR reports two potential products of different sizes/locations, one matching the sequence we amplified and analyzed, and another predicted by the in-silico tool at 1,997,769 + 2,000,086. Given the presence of repetitive sequences in non-coding sequences, we suspect either two predicted binding sites for one primer or some aspect of error in the genome contig may explain these two different predicted products.

Page 46: Table 2 “Notes” column reports the number of bps amplified for each peak. It is not clear what the meaning of this column is as when using the primers given in this table the amplified product is large than stated for all all peak. For example, the amplified product for peak 2 is as 434bps, however, the using the primers given the product size and based on the genome coordinates the product size should be 1271bps.

We again appreciate the diligent review of this information and meant to indicate the actual number of base pairs amplified, and we have fixed the error for peak2.

Page 9, line 1-4: “ peak 5 transgenic lines express EGFP within a subset of ventral Kolmer-Agdhur (KA) neurons in the spinal cord, which contact cerebrospinal-fluid, as identified by cell shape and localization. By 5 dpf, EGFP expression was consistently observed in KA neurons

across all lines, and in the heart,.....” Is sox10 expressed in KA neurons and cells within heart or is this thought to be ectopic activity of the peak 5 enhancer?

We were not able to identify definitively in the literature if KA neurons express sox10, but believe this is plausible given the expression of sox10 within some CNS structures. In terms of heart expression, the neural crest is believed to contribute to cardiac structures, so it is again plausible that this expression represents bona fide sox10 expression. We have clarified this point in the text (page 8-9), noting that ectopic expression remains a possibility.

Page 10, line 13: Control peaks1 and peak 8 are said to be embryonically expressed, does this exclude neural crest activity?

Visualizing these transgenics using the same scope and settings as the other reporter transgenics, we noted GFP for Peak1 in fin mesenchyme and for Peak8 in the CNS, but did not appreciate expression in cells located in or with morphology (e.g. stellate) consistent with the neural crest.

Reviewer 3 The objective of the study by Cunningham and colleagues is to identify mechanisms through which neural crest (NC) developmental transcriptional programs reemerge to promote melanoma. Previous work by this group (and others) showed that the NC gene *crestin* is expressed in pre-malignant melanoma nevi in zebrafish, which then progress to tumors. Such observations suggest that acquisition of an embryonic NC identity is important for tumor formation/progression. Here, the authors use ATAC-Seq to identify potential regulatory elements located adjacent to the *sox10* locus that are active in melanoma tumors. Transgenic approaches are then deployed to show a particular enhancer element (called peak-5) can specifically drive expression in NC cells and melanoma, and contains SoxE binding sites, suggesting a feed-forward mechanism maintains *sox10* expression during NC development and melanoma progression. Overall, the quality of the work is excellent and the authors have done a substantial amount of work to verify the location and activity of a regulatory elements at the *sox10* locus. However, the main conclusions of the study are not yet supported by data, as described below.

Major suggestion:

1) In the Abstract and Introduction, the authors describe the need to understand how epigenetic programs may be required to re-activate NC developmental programs to promote melanoma. However, this study is largely a standard “promoter-bashing” project that does not include analysis of epigenetic mechanisms, such as histone modifications or DNA methylation. Thus, the data does not fully support the authors claims concerning new knowledge on epigenetic regulation of NC transcriptional programs in melanoma progression. The epigenetic mechanisms that maintain peak-5 in an open conformation are not described. Modification of the text in the Abstract and Introduction are needed to better harmonize the objective of the study with the outcomes/conclusions presented.

We appreciate this critique as we used likely an overly broad notion of “epigenetic” regulation in the text. We have modified the text throughout the Abstract, Intro, and Discussion to more accurately reflect the analyses in this study as aiming to understand the “transcriptional” regulation of *sox10* from specific enhancer domains, rather than using the term “epigenetic”. We look forward to future studies that incorporate more epigenetic analyses of histone/DNA modifications to incorporate into our functional enhancer/promoter analysis as performed in this study.

2) In order to show that transcriptional programs, epigenetic states or genes, need to be reactivated to promote melanoma progression, one must first show that they are turned off/downregulated at some point during development, such as in differentiated melanocytes. There is a key piece of data missing from this study, showing that one or more of the open chromatin peaks, such as peak-5, is silenced during NC/pigment cell development and then re-activated when melanoma forms/progresses. Currently, the data support a model in which the peak-5 enhancer is important to maintain *sox10* expression throughout development, differentiation and tumorigenesis (because Sox10 is already established as being critical for all three stages). There is no data showing *sox10* expression reemerges, so this concept is not being directly tested and cannot be concluded from the current study.

It is true that we lack chromatin accessibility data in the non-transformed melanocyte state in this study, and this is the focus of ongoing efforts in our lab to collect this technically more challenging cell population (relative to the more abundant overgrown melanoma cells). Based on the observations of *crestin:EGFP* reporter expression in developing neural crest

which is then not seen in differentiated melanocytes but then is readily detected in melanoma cells (Kaufman et al, Science, 2016), we would favor the notion that, at least functionally as shown by crestin-driven EGFP as a read-out of neural crest progenitor identity and, similarly in this study, with peak5-driven EGFP, a neural crest progenitor program becomes more active (EGFP reporter becomes detectable) in the melanoma state. We have attempted to soften our statements/more precisely describe this in the text as an enforcement or enhancement of aspects of neural crest progenitor identity in melanoma formation rather than entirely “reactivated” as noted, as some level of *sox10* is needed for melanocyte development and potentially maintenance, as noted here.

Minor Suggestions:

1) The authors rely significantly on the use of the crestin marker and transgenic line. While originally described as a pan-NC marker, it preferentially labels cells at later NC progenitor stages (after induction) and trunk ectomesenchymal derivatives, such as pigment cells. Consistent with this notion, analysis of the *sox10* regulatory region in zebrafish by ATAC-Seq using an earlier NC progenitor marker (called *foxd3*- see Dev Cell., PMID 30513303) does not appear to identify the peak-5 enhancer, suggesting that the chromatin state identified in this study is more restricted to the pigment cell progenitor lineage, not early NC progenitors. The authors should directly show a comparison of the different ATAC-Seq data sets and, if appropriate, reconcile any differences in the Discussion.

We agree this is a fascinating line of inquiry in terms of determining a) which domains become accessible during which stages of neural crest formation, and b) which domains remain (or again become accessible or bound by specific transcription factors, see above) during melanoma formation. We reanalyzed our ATAC-Seq data in comparison with the studies suggested and others (Supp Fig 1B). As the reviewer astutely notes, the peak5 region and others become accessible at 5-6 somite stage but not earlier at bud and 75% epiboly in *foxd3*-expressing cells (from Lukoseviciute et al, Dev Cell, 2018). In addition, these regions also remain open in later *sox10*-positive cells at 16 and 20 somite stages (from Trinh et al, Cell Reports, 2017 and Ciarlo et al, eLife, 2017, respectively) as reported in two separate studies, strengthening this observation. We have updated the results (page 6) to note the timing of the appearance of the open domains during development, and highlight that the later appearance of the open domains near *sox10* in *foxd3*- and *sox10*-expressing embryonic cells by 5-6 somite stages is perhaps most consistent with NC progenitors that have already adopted a pigment cell progenitor identity in the discussion (page 18).

2) The resolution of many of the dark-field images is often too low to discern the NC cell labeling. The authors should include a higher magnification inset for each of the embryonic images to more clearly show which cells are labeled with the different transgenic constructs.

We have addressed this by including insets, as suggested for Fig 1d-f, and additionally with Zeiss lightsheet images for peak5:EGFP and *sox10*:mRFP visualizing colabeling in Fig 2d-e.

REVIEWERS' COMMENTS:

Reviewer #1 (Remarks to the Author):

In this revision, the authors have performed a number of additional experiments and textual changes that substantially improved the original study. They have successfully addressed my previous concerns.

Reviewer #2 (Remarks to the Author):

My critiques of the first submission have been adequately addressed. In particular the revised version of this manuscript has been substantially improved by the addition of experiments in which the peak5 enhancer has been deleted.

Minor issue

"Of note, the well-studied colourless mutant in the *sox10* gene shows only a reduction in *sox10* expression even in the homozygous state by WISH35, which would be consistent with the lack of apparent heterozygous phenotype in colourless as compared to our *sox10^{min}* mutation which leads to decreased *sox10* expression even in the heterozygous state and has a phenotype (see below). "

The logic in this paragraph is unclear. In two of three colourless alleles described in Dutton et al 2001 (ref 35) there is a frame shift and a premature stop. Given the transcript cannot make a functional protein, why is it relevant that there is "only a reduction in *sox10* expression." . It is clear the phenotype of the promoter deletion here is more penetrant, but the explanation for the difference is not clear. Transcriptional adaptation is a possibility; if this is the mechanism the authors are considering they should be explicit about it.

Reviewer #3 (Remarks to the Author):

The Authors have addressed my major concerns. Rod Stewart.

Rebuttal Letter

REVIEWERS' COMMENTS:

Reviewer #1 (Remarks to the Author):

In this revision, the authors have performed a number of additional experiments and textual changes that substantially improved the original study. They have successfully addressed my previous concerns.

We are grateful for the reviewer's comments.

Reviewer #2 (Remarks to the Author):

My critiques of the first submission have been adequately addressed. In particular the revised version of this manuscript has been substantially improved by the addition of experiments in which the peak5 enhancer has been deleted.

Minor issue

“Of note, the well-studied colourless mutant in the *sox10* gene shows only a reduction in *sox10* expression even in the homozygous state by WISH35, which would be consistent with the lack of apparent heterozygous phenotype in colourless as compared to our *sox10min* mutation which leads to decreased *sox10* expression even in the heterozygous state and has a phenotype (see below). “

The logic in this paragraph is unclear. In two of three colourless alleles described in Dutton et al 2001 (ref 35) there is a frame shift and a premature stop. Given the transcript cannot make a functional protein, why is it relevant that there is “only a reduction in *sox10* expression.” . It is clear the phenotype of the promoter deletion here is more penetrant, but the explanation for the difference is not clear. Transcriptional adaptation is a possibility; if this is the mechanism the authors are considering they should be explicit about it.

We are grateful for the reviewer's comments and have reworded the above sentence to more clearly describe our observation: “Interestingly, the well-studied *colourless* mutant in the *sox10* gene shows a reduction in *sox10* expression in the homozygous state by WISH and has no apparent heterozygous phenotype³⁵, compared to our *sox10min* promoter deletion mutation which leads to decreased *sox10* expression and a phenotype in the heterozygous state, as well as complete loss of *sox10* expression by WISH in homozygous mutants (see below).”

Reviewer #3 (Remarks to the Author):

The Authors have addressed my major concerns. Rod Stewart.

We are grateful for the reviewer's comments.